# Smarter Not Harder: Generative Process Evaluation with Intrinsic-Signal Driving and Ability-Adaptive Reward Shaping

**Tao He**[1*], **Rongchuan Mu**[1*], **Lizi Liao**[3], **Yixin Cao**[4], **Yang Li**[5], **Yijia Luo**[6],
**Weixun Wang**[6], **Ming Liu**[1,2†], **Bing Qin**[1,2]

[1] Harbin Institute of Technology, [2] Pengcheng Laboratory, [3] Singapore Management University
[4] Fudan University, [5] Shanghai Jiao Tong University, [6] Alibaba Group
the@ir.hit.edu.cn    rcmu@ir.hit.edu.cn
[*] Equal contribution.    [†] Corresponding author.

## Abstract

Large reasoning models (LRMs) have shown strong performance in complex mathematical reasoning when optimized via reinforcement learning (RL). However, conventional outcome-only reward provides sparse feedback, leading to inefficient optimization. In this work, we investigate whether generative process reward models (GenPRMs) can accelerate RL training of LRMs by improving the utilization of reasoning trajectories. We first analyze critical limitations in existing GenPRMs, including their heavy reliance on reasoning ability during correctness judgment, and suppression of exploration as well as vulnerability to reward hacking during reward assignment. To address these limitations, we first propose a novel **intrinsic-signal-driven evaluation** mechanism, which judges reasoning steps using semantic cues from the solution, thus mitigating extensive dependence on GenPRM. Furthermore, we (i) adopt **thought-level rewarding granularity** to alleviate over-dense step rewards, and (ii) design a **difficulty-aware reward formulation** that dynamically balances exploration and exploitation and keeping the optimization target of key tokens to mitigate reward hacking. We integrate these innovations into the process reward-based GRPO, resulting in the proposed **TP-GRPO** algorithm. Experiments on LRMs with 1.5B and 7B parameters show that TP-GRPO achieves higher improvements while using significantly fewer training samples, and more analyses further confirm the effectiveness of our proposed process evaluation mechanism.

## 1 Introduction

Reinforcement learning with verifiable rewards (RLVR) has recently achieved notable success in enhancing the reasoning capabilities of large reasoning models (LLMs), represented by DeepSeek R1 (Guo et al., 2025) and Kimi k1.5 (Team et al., 2025b). These approaches leverage efficient, rule-based outcome rewards (Shao et al., 2024; Yue et al., 2025; Xu et al., 2025) to extend the RL paradigm to complex reasoning tasks. However, existing RLVR methods typically focus solely on sparse answer correctness and overlook the rich semantic signals in intermediate reasoning trajectories. Motivated by the Metacognitive theories (Schraw & Moshman, 1995), outstanding students not only verify answers but also review and refine reasoning process, selectively reinforcing correct paradigms and correcting erroneous steps. This motivates researchers to explore **employing process rewards for better exploitation, thereby improving the training efficiency of LRM.**

Process reward models (PRMs) are designed to evaluate the quality of reasoning processes. Early discriminative PRMs (Lightman et al., 2023; Wang et al., 2023b; Zhang et al., 2025; Zheng et al., 2024) partly address the sparsity of outcome rewards but commonly suffer from unstable and subjective step segmentation, poor generalization, and high annotation costs (Guo et al., 2025). Generative PRMs (GenPRMs) have emerged as a more flexible alternative, using strong LRMs to assess reasoning traces via thinking (Khalifa et al., 2025; Feng et al., 2025; She et al., 2025; Liu et al., 2025; Chen et al., 2025). While promising, we find that naïvely applying GenPRMs to reward shaping can introduce critical pitfalls that substantially impact training stability and mislead optimization direction.

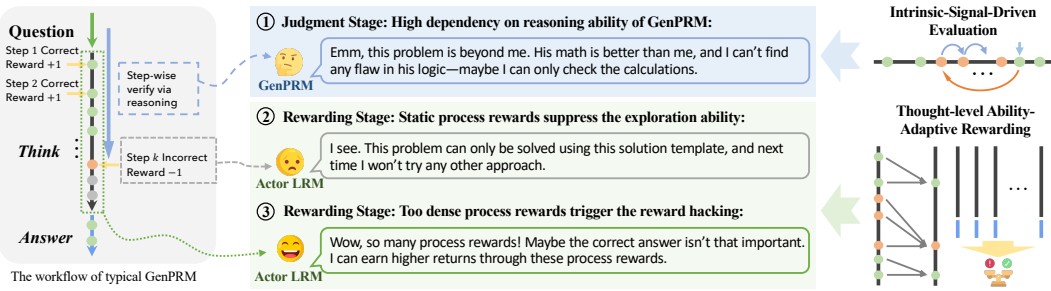

Figure 1: Illustration of the three target issues across two stages of the prior GenPRM workflow.

We divide the workflow of GenPRM into two stages: judging the correctness of each step; and then, assigning rewards for each step based on the evaluation results. Through empirical analysis, we identify critical challenges in both two stages. A clearer explanation of these issues can be found in Figure 1. In the judgment stage, existing GenPRMs primarily judge per-step correctness by re-deriving or simulating reasoning (Feng et al., 2025). Although improving generalization, it implicitly assumes that the reasoning capability of PRM exceeds that of actor LRM. This assumption not only raises the capacity requirements for GenPRM, but also undermines the evaluation reliability when confronted with hard tasks. Moreover, using reasoning to validate reasoning processes may also introduce potential bias, which becomes more pronounced in self-evolution settings where the LRM evaluates its own outputs (He et al., 2025). In the reward assignment stage, the common method of assigning static rewards (+1/-1) for dense steps risks suppressing exploration and triggering reward hacking. Penalizing mistakes during exploring challenging tasks may discourage beneficial trial-and-error behaviors. Besides, dense step rewards can dominate advantage estimation (Experimental results in Figure 5 verifies this issue), incentivizing the model to optimize for process profits maximization rather than final correctness, a trap known as reward hacking (Liu et al., 2024).

In response to these challenges, we first propose three guiding principles for GenPRM design: (P1) **Decouple evaluation from reasoning** to reduce high dependency on the reasoning ability of GenPRM; (P2) **Reward in an appropriate granularity** to mitigate the bias introduced by dense reward signals; and (P3) **Balance exploration and exploitation** to ensure that process rewards do not hinder exploratory reasoning. Guided by these principles, we introduce our GenPRM framework. First, in the evaluation stage, we propose an **Intrinsic-Signal-Driven Evaluation** mechanism. This strategy leverages semantic and logical signals inherent in the solution to judge step correctness. This design shifts the heavy reliance from the reasoning ability to PRM's semantic understanding and matching capacities, which constitute fundamental abilities of LLMs. Second, in the reward assignment stage, we design a **Thought-Level Difficulty-Aware Adaptive Reward** mechanism. Instead of rewarding each step, we merge consecutive correct or incorrect steps into coherent reasoning segments, which we refer to as "thought". Assigning rewards at the thought level effectively mitigates the issue of dense step rewards. Furthermore, the reward strength is dynamically adjusted according to the LRM's current reasoning capability: for difficult tasks, the mechanism automatically reduces the suppression for failed exploration; for easy tasks, more prominent process rewards are used to reinforce correct thoughts and penalize incorrect ones. In addition, our reward design keeps the optimization target of key tokens when introducing process reward, thereby effectively alleviating the issue of reward hacking. Finally, we integrate these mechanisms into the process reward-based GRPO (Shao et al., 2024) and propose our TP-GRPO algorithm.

We validate our approach on the DeepSeek-R1-Distill-Qwen model family (Guo et al., 2025). On AIME 2024 (Maxwell-Jia, 2024), our 1.5B model trained on only 700 problems achieves a +4.32% improvement, which rises to +5.98% when trained on 1,800 problems; the 7B model trained on 1,070 problems improves by +6.67%. While modest in absolute scale, these gains are consistent and achieved in fewer training steps, supporting our hypothesis that reasonable GenPRM can improve training efficiency beyond outcome-only rewards. Our main contributions are summarized as follows:

- An in-depth analysis of design pitfalls in GenPRM-based process evaluation, identifying three fundamental challenges and proposing three actionable principles for GenPRM design;
- Three innovations over challenges for existing GenPRM workflow are introduced: intrinsic-signal-driven evaluation, and thought-level, ability-adaptive reward schemes, further integrated with GRPO to form a new RL algorithm, termed TP-GRPO;

- Empirical evidence and detailed analysis show that TP-GRPO results in better efficiency and accuracy, underscoring the broader importance of well-designed GenPRM.

## 2 PRELIMINARY

### 2.1 PROBLEM FORMULATION

Given a mathematical problem $x$, the LRM $\pi_\theta$ produces a long-form, chain-of-thought output $o$, also termed Long CoT, which is formalized as the ordered pair comprising a ***think*** and an ***answer*** (Guo et al., 2025). The *think* encapsulates the full reasoning trajectory, typically spanning multiple rounds of hypothesis formation, exploratory derivation, error correction, and reflection. The *answer* $A$ is usually a concise summary based on the preceding *think*, generally retaining only the principal reasoning pathways that substantiate the final result. Assuming consistency between *answer* and *think*, we argue that improving reasoning quality primarily hinges on refining the *think*. Consequently, our process-level evaluation targets the *think* component rather than the *answer*. For brevity, unless otherwise specified, we will abuse the term solution to denote *think*.

### 2.2 RL FOUNDATION: PROCESS SUPERVISION GRPO

Our method builds on process supervision GRPO, first introduced in DeepSeekMath (Shao et al., 2024). Its optimization objective is identical to that of outcome supervision GRPO:

$$
\mathcal{J}_{\text{GRPO}}(\theta) = \mathbb{E}[q \sim P(Q), \{o_i\}_{i=1}^G \sim \pi_{\theta_{old}}(O|q)]
$$
$$
\frac{1}{G} \sum_{i=1}^G \frac{1}{|o_i|} \sum_{t=1}^{|o_i|} \left\{ \min \left[ \rho_{i,t} \hat{A}_{i,t}, \text{clip}(\rho_{i,t}, 1-\epsilon, 1+\epsilon) \hat{A}_{i,t} \right] - \beta D_{\text{KL}}[\pi_\theta \| \pi_{\text{ref}}] \right\}, \tag{1}
$$

where $\rho_{i,t}$ denotes the policy ratio, $\hat{A}_{i,t}$ the advantage estimate, $\epsilon$ the clipping threshold, $\beta$ the KL regularization coefficient, and $\pi_{\text{ref}}$ the reference policy. In process supervision GRPO, given $G$ sequences of process-reward signals $R = \{\{r_1^{\text{index}(1)}, \ldots, r_1^{\text{index}(K_1)}\}, \ldots, \{r_G^{\text{index}(1)}, \ldots, r_G^{\text{index}(K_G)}\}\}$, where $K_i$ represents the number of process rewards in $i$-th rollout, GRPO first standardize all process rewards:

$$
\hat{r}_i^{\text{index}(j)} = \frac{r_i^{\text{index}(j)} - \text{mean}(R)}{\text{std}(R)}. \tag{2}
$$

The per-token advantage is then defined as the cumulative sum of standardized process rewards at and after the token position:

$$
\hat{A}_{i,t} = \sum_{\text{index}(j) \geq t} \hat{r}_i^{\text{index}(j)}, \tag{3}
$$

where $\text{index}(j)$ denotes the token position in the $i$-th sequence at which the $j$-th process reward is assigned. This construction refines the advantage estimate using fine-grained, token-level signals.

## 3 OUR METHOD: TWO-STAGE THOUGHT-LEVEL GENPRM AND TP-GRPO

In this section, we first introduce a novel GenPRM mechanism, which primarily leverages intrinsic signals from the reasoning trajectories for judgment, and adaptively computes rewards to balance exploration and exploitation while mitigating the issue of reward hacking §(3.1). To assess its effectiveness, we integrate the designed GenPRM into the training loop of LRM to propose a new RL algorithm TP-GRPO §(3.2).

### 3.1 INTRINSIC SIGNAL-DRIVEN THOUGHT-LEVEL GENERATIVE PROCESS EVALUATION

Before evaluation, we segment the *think* into steps with appropriate semantic granularity. We compare two strategies: heuristic splitting based on "\n\n" and segmentation via prompting LRM. Finally, we adopt the method of prompting the LRM for good segmentation effectiveness. The applied prompt is presented in Appendix F.

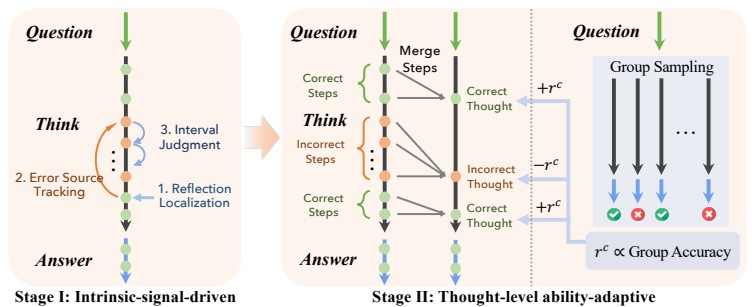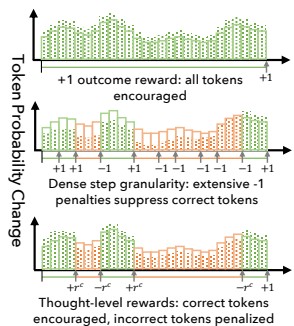

(a) Our proposed GenPRM method, consisting of: (i) decomposing correctness assessment into three easier sub-tasks, (ii) aggregating steps into thought and employing a difficulty-aware adaptive reward.

(b) Illustration of optimization bias induced by dense reward granularity.

Figure 2: Overview of our proposed GenPRM (a) as well as the illustration of optimization bias caused by dense process rewards (b). We design distinct evaluation and reward mechanisms for correct and incorrect solutions; the overview here illustrates only the pipeline for correct solutions.

Building on this decomposition, we propose our GenPRM framework, which can be broadly divided into two stages: correctness evaluation and reward assignment. In Stage I, we design a prompt pipeline to judge per-step correctness. Our method leverages semantic and logical signals contained in the solution, thereby reducing the dependency on strong reasoning capacity of GenPRM. In Stage II, we introduce a thought-level, difficulty-aware adaptive process reward formulation. This mechanism dynamically balances exploration and exploitation and mitigates reward hacking by keeping the optimization target of key tokens. The overview of our proposed GenPRM is given in Figure 2a. A more detailed illustration is presented by Figure 7 in Appendix F.

### 3.1.1 STAGE I: INTRINSIC-SIGNAL-DRIVEN STEP-LEVEL PROCESS EVALUATION

In this subsection, we initiate our exploration of employing a generative paradigm to judge the correctness of reasoning steps. Our process evaluation mechanism is designed around the core principle: **Decoupling Evaluation from Reasoning Capacity.**

This principle is proposed to solve **the 1st Pitfall** of existing GenPRM paradigm: The quality of current process evaluation mechanisms is fundamentally constrained by the reasoning capabilities of the GenPRM. This requires the GenPRM to identify logical errors within the solution through deliberate thinking, thereby implicitly assuming that GenPRM's reasoning proficiency is at least on par with the actor LRM. However, as task complexity increases and the actor LRM's reasoning capability continues to advance, the demand on the GenPRM escalates accordingly, which poses a serious challenge to the long-term effectiveness of the GenPRM as an evaluator.

To solve this challenge, we suggest decoupling evaluation from reasoning by breaking down the evaluation task into sub-tasks completed by other foundational capabilities of LLMs.

By examining Long CoT, we observe intrinsic semantic cues indicative of the LRM's reasoning state, such as self-reflective statements. These reflections directly indicate errors and their potential causes, thereby providing instructive signals for process evaluation. Building on the design principle and these findings, we design two distinct protocols for correct and incorrect solutions:

- **Correct Solutions (Identify Effective Steps):** For traces with correct final answers, we pinpoint ineffective steps via three steps: 1) *Reflection Localization*: Using semantic comprehension, GenPRM identifies reasoning steps where the LRM realizes its mistakes through self-reflection (e.g., "wait, I made a mistake"). 2) *Error Source Tracking*: During reflection, the LRM typically analyzes the causes of the reflected mistake. Based on the analysis, GenPRM traces back error source and delineates a candidate error interval. 3) *Interval Verification*: GenPRM applies heuristic rules (e.g., "steps dependent on conclusions from incorrect previous steps are deemed erroneous") to judge each step within the error interval, guided by the reflected error cause.

- **Incorrect Solutions (Avoid Over-Penalization):** For incorrect solutions, we first propose a conservative hypothesis: all steps in the *answer* are erroneous. Under this hypothesis, *think* steps

semantically matching the erroneous *answer* are labeled incorrect, while unmatched steps are viewed as uncertain. **This design realizes a trade-off:** relying solely on the outcome rewards would incur over penalization for underthinking behaviors in the *think* part, but achieving perfect step-level judgments demands exceptionally high evaluation capability for GenPRM. We propose an easy process evaluation requiring only the capacity of semantic alignment for GenPRM.

This evaluation framework is grounded in the following theoretical and practical motivations:

- For correct solutions, we leverage reflection signals to trace erroneous steps, based on the **self-contained assumption**: if the final answer is correct, any erroneous reasoning step must be accompanied by an effective reflection that identifies and corrects the error (either by direct fix or switching reasoning strategies). Otherwise, the error would propagate and mislead the final answer. Under this assumption, errors and reflections must occur in pairs in correct solutions, enabling a reflection-first attribution strategy that is both theoretically valid and practically reliable.

- We further emphasize decoupling evaluation from reasoning. For correct solutions, we perform evaluation using intrinsic reflection cues combined with GenPRM's semantic understanding capability; for incorrect solutions, we conduct coarse-grained evaluation based solely on semantic matching. This design substantially reduces reliance on GenPRM's reasoning capabilities. There are two core motivations: (1) Semantic understanding and matching are more fundamental and stable abilities of LLMs, leading to more robust evaluation results; (2) requiring GenPRM to conduct process evaluation via thinking of inconsistencies between its own reasoning chain and the Long CoT implicitly presupposes stronger and heterogeneous reasoning capabilities. If such capabilities are not satisfied, the evaluation process is likely to introduce systematic bias. By decoupling evaluation from reasoning, we effectively mitigate such risks.

### 3.1.2 STAGE II: THOUGHT-LEVEL ABILITY-ADAPTIVE PROCESS REWARDING MECHANISM

Specific reward values are required after per-step correctness evaluation.

**S1: Thought-level Reward Unit through Step Merging**

**The 2nd Pitfall:** Through analyzing the step-level process evaluation results, we observe that *think* is often decomposed into a large number of steps. If process rewards are assigned at the step level, the advantage would be dominated by these process rewards, which could in turn mislead the optimization to incorrect directions. Figure 2b illustrates this effect: although the 4th step is correct, the five subsequent incorrect steps result in a negative cumulative return. Consequently, this correct step is erroneously penalized during training, highlighting the misleading nature of dense process rewards.

To address the aforementioned issue, we revisit the granularity of reward assignment and propose a thought-level process reward mechanism. Consecutive steps with the same correctness are merged into a single logical unit—termed a thought. For correct solutions, this entails merging **consecutive** correct/incorrect steps into a correct/incorrect thought; for incorrect solutions, consecutive matching or non-matching steps are merged analogously. The process reward is then assigned at the thought level rather than the step level. This design is easy to implement and effectively reduces the density of process rewards. Importantly, we do not blindly filter the reward signals; instead, we retain the minimal set of rewards necessary to correctly guide LRM optimization while maximizing the reduction of redundant rewards. A more intuitive comparison is presented in Figure 2b.

**S2: Difficulty-Aware Adaptive Process Reward Mechanism**

We further design two distinct process reward mechanisms based on the solution correctness. We argue that designing effective process rewards for LRMs requires addressing **two major Pitfalls**:

- Static correctness-based process rewards can unintentionally suppress exploration. Although penalties for incorrect attempts lower the probability of unproductive trials, they may also restrict LRM's exploratory capabilities. Process rewards provide more localized learning signals that bias LRM toward exploiting known high-return reasoning paths. However, this also increases the risk of falling into local optima and diminishes further exploration.

- Poorly designed process rewards may introduce reward hacking—the model may discover ways to improve the process rewards without genuinely improving its reasoning.

To this end, we propose a difficulty-aware adaptive process reward mechanism with two complementary regimes, defined according to the correctness of solutions.

- **For Correct Solutions:** For a correct solution $o_i$, each correct thought receives $+r^c$, and each incorrect thought receives $-r^c$, where:

$$r^c = \alpha \cdot \text{acc}_G, \quad \alpha > 0.$$

  Here, $\text{acc}_G$ is the accuracy of the $G$ sampled solutions for the same problem. When $\text{acc}_G = 0$, $r^c$ vanishes, reducing to outcome supervision GRPO (Guo et al., 2025) and prioritizing exploration. When $\text{acc}_G = 1$, $r^c = \alpha$, process rewards are strongest to reinforce correct steps and suppress incorrect ones. The hyperparameter $\alpha$ further controls the global scale of the process rewards. By scaling $r^c$ with accuracy, the model is free to explore facing harder questions while receiving stronger process guidance once the task is easy, thus avoiding both premature suppression of exploration and spurious optimization behaviors.

- **For Incorrect Solutions:** For an incorrect solution $o_i$, penalties are applied only to thoughts that lead to the erroneous answer, i.e., those in the *think* matched to the *answer* part. Given the normalized outcome reward $\hat{r}_i^o$ in Eq. 4, the thought-level process reward $r_i^{ic}$ is defined as:

$$r_i^{ic} = \begin{cases} \hat{r}_i^o, & \text{if the } i\text{-th thought matches the incorrect } answer, \\ -\hat{r}_i^o, & \text{otherwise.} \end{cases}$$

  Since the solution is incorrect and $\hat{r}_i^o \leq 0$, unmatched thoughts, typically other exploratory attempts that are not employed in the *answer* part, receive a non-negative reward. This prevents penalizing uncertain reasoning steps, thereby mitigating the risk of over-penalization.

We further discuss how our method contributes to mitigating reward hacking in the following section.

## 3.2 TP-GRPO: GRPO based on Thought-level Process Reward

Finally, we employ the process evaluation results in process supervision GRPO (Shao et al., 2024) and introduce a new algorithm, TP-GRPO. TP-GRPO jointly optimizes the LLM under the guidance of both process and outcome rewards. For the outcome rewards, we begin by normalizing them within each group as in outcome supervision GRPO (Guo et al., 2025):

$$\hat{r}_i^o = \frac{r_i^o - \text{mean}(\{r_j^o\}_{j=1}^G)}{\text{std}(\{r_j^o\}_{j=1}^G)}. \tag{4}$$

The process rewards are configured as described in Sec 3.1.2, without applying the normalization as in Eq. 2. We explain this design in Appendix C. Following Shao et al. (2024), the advantage of each token $o_{i,t}$ is calculated as $A_{i,t} = \sum_{t'=t}^{|o_i|} r_{i,t'}$ as introduced in Eq 3, where $|o_i|$ is the number of tokens in $o_i$. Finally, we optimize the LRM following the optimization objective defined in Eq. 1:

To better understand the advantages of the process rewards design, we establish the following theoretical properties from the perspective of advantage estimation in GRPO:

**Proposition 1.** *For a correct solution $o_i$, token $o_{i,t}$ in correct thoughts have advantage $A_{i,t} = \hat{r}_i^o$; while token $o_{i,t}$ in incorrect thoughts have advantage $A_{i,t} = \hat{r}_i^o - r^c$.*

*Remark:* This proposition shows that tokens within correct thoughts retain the same advantage as using outcome-only rewards, safeguarding against misguided shifts in the optimization objective during RL training — which we posit as the primary cause of reward hacking. In contrast, tokens within incorrect thoughts receive reduced advantages compared with using outcome-only rewards, effectively weakening the optimization of flawed steps.

**Proposition 2.** *For an incorrect solution $o_i$, token $c_t$ in matched thoughts have advantage $A_{i,t} = \hat{r}_i^o$, while token $c_t$ in unmatched thoughts have advantage $A_{i,t} = 0$.*

*Remark:* This proposition demonstrates that tokens within matched thoughts retain the same advantage values as under outcome-only rewards, thereby achieving the objective mentioned above: introducing process rewards without altering the original training objective for these incorrect tokens during RL training. In contrast, unmatched thoughts receive zero advantages, avoiding over-penalization since their correctness remains uncertain.

The proofs of these two propositions are presented in Appendix B.

## 4 EXPERIMENTS

In this section, we evaluate TP-GRPO on five datasets to investigate the following research questions:

- **RQ1:** How does TP-GRPO perform in enhancing the training efficiency of LRM? (§ 4.2)
- **RQ2:** Does intrinsic-signal-driven step-level process evaluation provide advantages over the reasoning-based GenPRM paradigm? (§ 4.3.1 & § 4.3.3)
- **RQ3:** Does dense reward design introduce potential negative effects, and is the proposed step-merge strategy really effective? (§ 4.3.1)
- **RQ4:** Is the proposed difficulty-aware reward mechanism effective in practice? (§ 4.3.1)
- **RQ5:** Are the process rewards designed for correct and incorrect solutions empirically rational? (§ 4.3.2)

Given computational limitations, we focus our experiments on smaller LRMs. To meet the format requirement that explicitly includes both *think* and *answer* components, we select the DeepSeek-R1-Distill-Qwen 1.5B and 7B for experiments. We leave broader studies across more scales and variants to future work.

### 4.1 EXPERIMENTAL SETTINGS

**Baselines**   We primarily compare TP-GRPO with the reproduction of GRPO using only outcome rewards, aiming to demonstrate the advantage of our proposed process rewards in terms of both performance and efficiency. In addition, we seek to validate the effectiveness of our approach relative to the existing GenPRM methods. To this end, we include two GenPRM-based baselines: (i) an LLM-as-a-judge variant based on Qwen3-32B that evaluates step correctness using prompts derived from Zhao et al. (2025), and (ii) GenPRM-32B model that is pretrained on process evaluation datasets (Zhao et al., 2025). The prompt used for LLM-as-a-judge is provided in Table 16 at the end of the Appendix. Besides, we also include results from several open-source baselines trained on DeepSeek-R1-Distill-Qwen models as reference and also compare efficiency with them.

**Training Data and Evaluation Benchmark**   We conduct all experiments on DeepScaler-40K (Luo et al., 2025). We follow prior work and evaluate on 5 widely-used math benchmarks: AIME 24 (Maxwell-Jia, 2024), AIME 25 (math ai, 2025), AMC 23 (Li et al., 2024a), MATH-500 (Hendrycks et al., 2021), and Olympiad (He et al., 2024). For experiments with DeepSeek-R1-Distill-1.5B, we perform 16 independent Pass@1 evaluations and report the average (Avg@16) as final results. Due to the higher inference cost of DeepSeek-R1-Distill-7B and limited sizes of AIME 24, AIME 25, and AMC 23, we report Avg@5 on these subsets and Pass@1 on MATH-500 and Olympiad. Besides, we aim to further investigate improvements in training efficiency. To ensure a fair comparison, we design a new metric **Effic.** to measure the training efficiency: $\textit{\textbf{Effic.}} = \frac{\text{Improvement}}{\text{\#training solutions}} * 10^5$. Here, "Improvement" denotes the reasoning performance gain, and "#solutions" is the total number of solutions used for training. Obviously, a higher *Effic.* indicates that the algorithm can achieve greater performance with fewer samples, thereby reflecting higher training efficiency.

**Hyperparameters**   All training experiments are conducted using the TRL framework (von Werra et al., 2020) and vLLM backend. We use prompt batch size=5, a fixed learning rate=$1e-6$, and sample 8 rollouts per prompt. For on-policy GRPO, we save checkpoints every 50 training steps and perform evaluations accordingly. Owing to the efficiency constraints of GenPRM, we train the framework in an off-policy setting. Each iteration consists of: (i) collect rollouts using the latest model sufficient for 50 training steps, (ii) parallel evaluation deploying multiple GenPRMs, and (iii) performing multi-step training once evaluation is complete. More details about off-policy pipeline are presented in Appendix G. Further details of the parameter settings are relegated to the Appendix E.

### 4.2 MAIN RESULTS

Tables 1 and 2 report the main results with DeepSeek-R1-Distill-Qwen-1.5B and 7B as backbone models, respectively.

- **Absolute Performance Improvements.** On the 1.5B model, TP-GRPO achieves an average improvement of +2.04, with substantial gains on AIME 24 (+4.32) and AIME 25 (+3.13). For

Table 1: Main results on DeepSeek-R1-Distill-Qwen-1.5B. TP-GRPO achieves higher accuracy than the on-policy outcome-based GRPO baseline while requiring $5\times$ fewer solutions. "K" means $\times 1000$.

| Model | AIME 24 Avg@16 | AIME 25 Avg@16 | AMC 23 Avg@16 | MATH-500 Avg@16 | Olympiad Avg@16 | Avg. | #Solution | *Effic.* |
|---|---|---|---|---|---|---|---|---|
| Base Model | 28.80 | 22.50 | 62.90 | 82.80 | 43.30 | 48.06 | - | - |
| *Outcome Reward-based RLVR methods* | | | | | | | | |
| AutoThink (Stage 1) | 28.13 | 19.58 | 60.39 | 79.74 | 40.44 | 45.66 | 450.6K | -0.53 |
| Open-RS1 | 30.42 | 20.83 | 63.78 | 83.01 | 43.91 | 48.39 | 57.6K | 0.53 |
| Still-3-1.5B-Previous | 32.50 | 23.33 | 66.94 | 84.19 | 45.96 | 50.58 | 256K | 0.57 |
| AdaptThink | 33.75 | 24.38 | 69.35 | 84.18 | 45.04 | 51.34 | 614.4K | 0.98 |
| AReaL-1.5B-Preview(Stage 1) | 30.83 | 23.75 | 66.64 | 84.33 | 45.51 | 50.21 | 1638.4K | 0.13 |
| DeepScaler-1.5B-Preview | 43.10 | 29.38 | 73.60 | 87.80 | 50.00 | 56.78 | >3584K | <0.49 |
| GRPO Replication (850 steps) | 32.71 | 24.58 | 64.53 | 82.70 | 43.67 | 49.64 | 34K | 4.65 |
| *Process Reward-based methods* | | | | | | | | |
| GRPO with LLM-as-a-judge (118 steps) | 30.41 | 24.58 | 63.32 | 82.61 | 43.35 | 48.85 | 4.7K | 16.8 |
| GRPO with GenPRM-32B (262 steps) | 31.45 | 23.12 | 63.00 | 83.30 | 43.40 | 48.86 | 10.4K | 7.63 |
| TP-GRPO (140 steps) | 33.12 | 25.63 | 64.01 | 83.81 | 43.91 | 50.10 | 5.6K | 36.43 |

Table 2: Main results on DeepSeek-R1-Distill-Qwen-7B. TP-GRPO achieves higher accuracy than the on-policy outcome-based GRPO baseline while requiring $1\times$ fewer solutions. "K" means $\times 1000$.

| Model | AIME 24 Avg@5 | AIME 25 Avg@5 | AMC 23 Avg@5 | MATH-500 Pass@1 | Olympiad Pass@1 | Avg. | #Solution | *Effic.* |
|---|---|---|---|---|---|---|---|---|
| Base Model | 54.00 | 36.67 | 81.20 | 92.00 | 55.11 | 63.80 | - | - |
| *Outcome Reward-based RLVR methods* | | | | | | | | |
| AutoThink (Stage 1) | 52.67 | 32.67 | 77.59 | 90.00 | 54.07 | 61.40 | 450.6K | -0.53 |
| ReasonFlux-F1 | 54.67 | 34.00 | 80.00 | 91.60 | 55.26 | 63.11 | 60K | -1.15 |
| AdaptThink | 52.67 | 38.00 | 82.17 | 90.80 | 56.89 | 64.11 | 307.2K | 0.10 |
| AReaL-boba-RL-7B | 60.00 | 44.67 | 86.02 | 93.60 | 60.89 | 69.04 | 2048K | 0.37 |
| Skywork-OR1-7B | 69.33 | 51.33 | 87.71 | 93.40 | 61.33 | 72.62 | >5406.7K | <0.26 |
| GRPO Replication (400 steps) | 58.67 | 38.00 | 81.20 | 92.40 | 56.44 | 65.34 | 16K | 9.6 |
| *Process Reward-based methods* | | | | | | | | |
| TP-GRPO (214 steps) | 60.67 | 43.33 | 83.13 | 92.00 | 57.04 | 67.23 | 8.56K | 40.07 |

7B model, the gain rises to +3.43, including +6.67 on AIME 24 and +6.66 on AIME 25. These consistent improvements over backbones validate the effectiveness of our process reward design.

- **Learning Efficiency over On-Policy GRPO.** TP-GRPO also outperforms the on-policy GRPO baseline that relies solely on outcome rewards. On the 1.5B model, TP-GRPO surpasses on-policy GRPO trained with 34K solutions (850 steps) using no more than 5.6K solutions (140 steps). Similarly, on the 7B model, TP-GRPO outperforms a 16K-solution (400-step) on-policy GRPO run with only 8.56K solutions (214 steps). This highlights a notable improvement in learning efficiency enabled by the proposed process reward mechanism.

- **Comparison with Other GenPRM Methods.** On the 1.5B LRM, we compare our method with two other GenPRMs. Although they achieve slight improvements over the base model, their performance remains far below TP-GRPO. This indicates that process rewards in these methods were not effectively leveraged, likely due to those pitfalls we discuss in the Method section, further highlighting the necessity of designing an appropriate process reward mechanism.

- **Comparison of Efficiency with Other Methods.** Against other well-performing approaches (e.g., DeepScaler-1.5B-Preview and Skywork-OR1-7B), TP-GRPO demonstrates faster performance gains with substantially fewer training samples, despite these models achieving higher final accuracy through much larger budgets. While such comparisons are inherently imperfect due to the non-linear convergence patterns of LM training, our results still show that TP-GRPO matches or exceeds competitive baselines with significantly less data.

- **Training Dynamics.** Figure 3 further illustrates the Avg@16 curves on AIME 24 for 1.5B experiments. Here, "off-policy GRPO" denotes off-policy GRPO variant without process rewards. All baselines are evaluated every 50 training steps. For TP-GRPO, we focus on the impact of process rewards; therefore, only solutions with non-zero process rewards are retained. As a result, the actual number of training steps per iteration is fewer than 50. Despite this, TP-GRPO's curve

is noticeably steeper, indicating higher training efficiency. This provides strong evidence that process rewards enhance the exploitation of reasoning trajectories for LRM training.

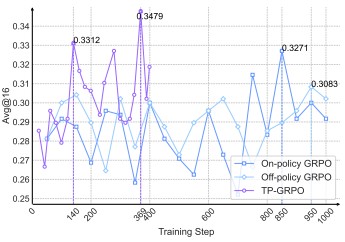

Figure 3: The trend of accuracy (Avg@16) on AIME 2024 over training steps for DeepSeek-R1-Distill-Qwen-1.5B experiments.

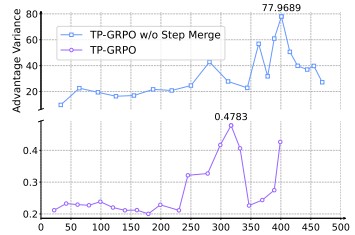

Figure 4: Advantage variance comparison between TP-GRPO and the variant w/o Step Merge.

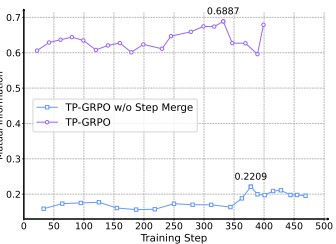

Figure 5: Mutual information between advantage and token correctness for TP-GRPO and its variant w/o Step Merge.

### 4.3 ANALYSIS

We also conduct more detailed analyses below. Due to page limitations, more analyses are included in Appendix H.

#### 4.3.1 ABLATION STUDY FOR PROPOSED STRATEGIES

We conduct ablation studies to assess the contribution of our proposed three strategies. For the intrinsic-signal-driven evaluation method, we replace it with direct judgment via LLM-as-a-judge. Detailed prompts are presented in Appendix. For the step-merge strategy, we omit merging and assign rewards directly at the step level. For the ability-adaptive process reward mechanism, we fix reward values to +1 for correct or uncertain steps and -1 for incorrect steps. Table 3 shows that all three strategies yield clear improvements, confirming the necessity of addressing their respective target issues. Notably, the step-merge strategy yields the largest improvement, despite its remarkably simple implementation. To investigate this reason, we analyze token-level advantage values: (i) the variance across tokens within the same rollout, higher variance implies divergent optimization targets, leading to instability; and (ii) the mutual information between token advantages and the correctness of the corresponding step/thought (simplified as token correctness), correct tokens should have higher advantages. Figure 4 and 5 show that step-level rewards cause large variance and low mutual information. **These outcomes arise as dense reward assignment enables consecutive incorrect steps to exert substantial negative impacts, which in turn distort, and even flip, the magnitude and sign of advantages in prior steps, thereby misleading the original optimization direction.** By merging consecutive steps with the same correctness and assigning rewards at the thought level, this issue is effectively mitigated.

#### 4.3.2 ABLATION STUDY FOR REWARDS

We design two distinct reward mechanisms for correct and incorrect solutions. To better understand their individual contributions, we conduct ablation studies under two settings: one using only process rewards from correct solutions, and the other using those only from incorrect solutions. Results are presented in Table 4. We observe an interesting phenomenon: for AIME 24, using only correct-solution rewards yields better performance, whereas on AIME 25, the opposite holds. This contrast exactly offers a compelling illustration of the distinct functions underlying the two reward schemes. Relying exclusively on rewards from correct solutions fails to mitigate the over-penalization for valid attempts in incorrect solutions, thereby constraining the model's exploratory capacity and ultimately yielding weak performance on the more challenging task AIME 25. In contrast, focusing exclusively on rewards from incorrect answers precludes the targeted reinforcement of useful patterns within correct solutions, leading to fewer gains on both AIME 24 and AIME 25 relative to the full TP-GRPO framework. In general, these results show that the proposed process reward mechanism supports effective learning from both correct and incorrect reasoning paths.

Table 3: Ablation study on improvement strategies. 'Stage I' is the intrinsic-signal-driven evaluation. 'S1' is step merge strategy, 'S2' refers to difficulty-aware adaptive reward formulation.

| Model | AIME 24 | AIME 25 | AMC 23 |
|---|---|---|---|
| TP-GRPO | 33.12 | 25.63 | 64.01 |
| - w/o Stage I | 31.04 | 23.54 | 63.93 |
| - w/o Stage II/S1 | 31.66 | 22.29 | 62.19 |
| - w/o Stage II/S2 | 32.71 | 22.92 | 63.47 |

Table 4: Ablation study on process rewards. 'CS reward' means 'reward for **C**orrect **S**olutions', while 'IS' means '**I**ncorrect **S**olutions'.

| Model | AIME 24 | AIME 25 |
|---|---|---|
| Base Model | 28.80 | 22.50 |
| TP-GRPO | 33.12 | 25.63 |
| - w/o CS Reward | 30.00 | 25.00 |
| - w/o IS Reward | 32.01 | 22.29 |

### 4.3.3 DIFFERENT LRMS FOR PROCESS EVALUATOR

To evaluate the dependency of our framework on the reasoning capabilities of the PRM, we further conduct an analysis study. We first conduct experiments using different LLMs as the generative PRM, and then we implement GenPRM using the LLM-as-a-Judge paradigm with these LLMs. We select Qwen3-32B, Qwen3-4B and Gemma-3-12B Team et al. (2025a), whose reasoning abilities on GPQA Diamond (Rein et al., 2024) are 65.6%, 55.9%, and 40.9%, respectively[1]. We applied these three GenPRMs to both TP-GRPO and LLM-as-a-Judge, with results shown in Table 5 and Figure 6. Each entry is separated by "/", with the left value for TP-GRPO and the right for LLM-as-a-Judge. The results indicate that despite substantial differences in reasoning ability, all GenPRMs consistently improve LRM performance through TP-GRPO. Although performance slightly decreases as the evaluator's reasoning ability declines, the drop is minimal. In contrast, LLM-as-a-Judge consistently underperforms TP-GRPO, with a pronounced decline. Results based on Gemma-3-12B-it are notably worse than those

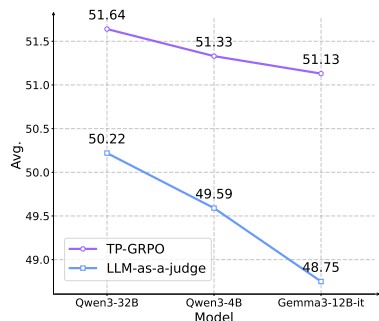

Figure 6: Performance trend when using different LLMs as evaluators. The x-axis shows different evaluators, whose reasoning ability gradually decreases from left to right.

based on Qwen3-32B and even below the base model. This suggests that existing generative process evaluation methods heavily rely on external GenPRM, especially on reasoning ability. Moreover, TP-GRPO consistently delivers stable improvements across different LLM scales (Qwen3-32B vs. Qwen3-4B) and families (Qwen3 vs. Gemma), further validating the effectiveness and robustness of our proposed process evaluation mechanism.

Table 5: Robustness Analysis with Different LLMs as GenPRMs.

| Model | AIME 24 | AIME 25 | AMC 23 | MATH-500 | Avg. | Step |
|---|---|---|---|---|---|---|
| Base Model | 28.80 | 22.50 | 62.90 | 82.80 | 49.25 | - |
| TP-GRPO / LLM-as-a-Judge | | | | | | |
| - Qwen3-32B | 33.12 / 30.41 | 25.63 / 24.58 | 64.01 / 63.28 | 83.81 / 82.61 | 51.64 / 50.22 | 140 / 118 |
| - Qwen3-4B | 33.33 / 29.79 | 24.58 / 23.54 | 63.78 / 63.10 | 83.62 / 81.94 | 51.33 / 49.59 | 155 / 122 |
| - Gemma-3-12b-it | 32.71 / 30.20 | 23.33 / 22.50 | 64.76 / 61.25 | 83.73 / 81.28 | 51.13 / 48.75 | 125 / 226 |

## 5 CONCLUSION

This work revisits the design of GenPRM and proposes two fundamental challenges: an over-reliance on reasoning capabilities and a suppression of exploratory behavior. To address these, we distilled a set of guiding principles for reward model design, leading to a robust evaluation mechanism driven by intrinsic signals and a thought-level difficulty-aware adaptive reward formulation that dynamically balances exploration and exploitation. Integrating these components with process supervision GRPO, our proposed TP-GRPO algorithm achieves substantial performance gains over an outcome-only RLVR baseline, even under fewer training steps. Our results demonstrate the efficacy of process reward modeling for improving optimization efficiency for LRMs. Beyond the empirical findings, this study raises broader questions about how to design reward signals that encourage both accurate and diverse reasoning processes. We hope our work serves as a foundation for developing more effective and generalizable GenPRM paradigms.

---

[1]Data from: https://www.datalearner.com/ai-models

## 6 ACKNOWLEDGMENTS

The research in this article is supported by the National Science Foundation of China (U22B2059, 62276083), Key Research and Development Program of Heilongjiang Province (2022ZX01A28) and the 5G Application Innovation Joint Research Institute's Project (A003).

## 7 ETHICS STATEMENT

This work does not involve human subjects, personal or sensitive data, or experiments that could raise ethical concerns. No new datasets containing private information were collected or released. The methods, analyses, and applications presented do not have foreseeable negative societal impacts, nor do they pose risks to privacy, security, or fairness. We have no conflicts of interest, sponsorship-related influence, or legal compliance issues to disclose. All research activities adhered to standard academic integrity practices, and no ethics committee approval was required for this study.

## 8 REPRODUCIBILITY STATEMENT

### 8.1 IMPLEMENTATION

Our implementation is based on TRL (von Werra et al., 2020), and the complete code is provided in the Supplementary Materials. The training dataset used is the public DeepScaler-40K dataset, which can be found in huggingface requires no preprocessing. For evaluation, we adopt publicly available benchmark datasets, including AIME24, AIME25, AMC23, MATH-500, and OlympiadBench.

### 8.2 EXPLANATIONS FOR TWO PROPOSITIONS IN SECTION 3.2

Detailed proofs are provided in Appendix B.

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

## A  THE USE OF LARGE LANGUAGE MODELS (LLMS)

For this paper, LLMs were employed exclusively for textual polishing, without contributing to conceptual ideation.

## B   PROOF OF PROPOSITION

**Proposition 1.** *For a correct solution $o_i$, token $o_{i,t}$ in correct thoughts have advantage $A_{i,t} = \hat{r}_i^o$; while token $o_{i,t}$ in incorrect thoughts have advantage $A_{i,t} = \hat{r}_i^o - r^c$.*

*Proof.* The proof is straightforward. After merging steps of the same type, the original *think* can be decomposed into an alternating sequence of correct and incorrect thoughts:

$$\texttt{think} = \text{thought}_1^{\text{correct}} \circ \text{thought}_2^{\text{incorrect}} \circ \cdots \circ \text{thought}_{n-1}^{\text{incorrect}} \circ \text{thought}_n^{\text{correct}}.$$

Accordingly, the thought-level reward sequence is:

$$\left[ r_{\text{index}(1)}, r_{\text{index}(2)}, \ldots, r_{\text{index}(n)} \right] = \left[ r^c, -r^c, \ldots, -r^c, \hat{r}^o \right],$$

where $\text{index}(j)$ denotes the index of the last token in the $j$-th thought. The last thought $t_n^{\text{correct}}$ receives only the outcome reward $\hat{r}^o$.

For tokens in a correct thought $\text{thought}_j^{\text{correct}}$, the advantage value is computed as:

$$A_{\text{index}(j-1)<k\leq\text{index}(j)}^{\text{correct}} = r_{\text{index}(j)} + r_{\text{index}(j+1)} + \cdots + r_{\text{index}(n)}. \tag{5}$$

Given the alternating reward pattern, this simplifies to:

$$\underbrace{r^c + \cdots + r^c}_{(n-j)/2 \text{ terms}} + \underbrace{(-r^c) + \cdots + (-r^c)}_{(n-j)/2 \text{ terms}} + \hat{r}^o = \hat{r}^o.$$

Similarly, for tokens in the incorrect thought $\text{thought}_i^{\text{incorrect}}$, the corresponding advantage is:

$$\begin{aligned}
A_{\text{index}(j-1)<k\leq\text{index}(j)}^{\text{correct}} &= r_{\text{index}(j)} + r_{\text{index}(j+1)} + \cdots + r_{\text{index}}(n) \\
&= \underbrace{r^c + \cdots + r^c}_{(n-j+1)/2-1 \text{ times}} + \underbrace{(-r^c - \cdots - r^c)}_{(n-j+1)/2 \text{ times}} + \hat{r}^o \\
&= \hat{r}^o - r^c
\end{aligned} \tag{6}$$

$\square$

**Proposition 2.** *For an incorrect solution $o_i$, token $c_t$ in matched thoughts have advantage $A_{i,t} = \hat{r}_i^o$, while token $c_t$ in unmatched thoughts have advantage $A_{i,t} = 0$.*

*Proof.* Following the same reasoning as in Proposition 1, for an incorrect *think*, we first merge steps of the same type and decompose the sequence into alternating matched and unmatched thoughts with respect to the *answer*. We assign a penalty to matched thoughts with reward $r^{\text{incorrect}} = \hat{r}^o$, and reward unmatched thoughts with $-r^{\text{incorrect}} = -\hat{r}^o$.

For a matched thought $t_j^{\text{matched}}$, the advantage of its tokens is computed as:

$$\begin{aligned}
A_{\text{index}(j-1)<k\leq\text{index}(j)}^{\text{matched}} &= r_{\text{index}(j)} + r_{\text{index}(j+1)} + \cdots + r_{\text{index}(n)} \\
&= \underbrace{\hat{r}^o + \cdots + \hat{r}^o}_{(n-j)/2 \text{ terms}} + \underbrace{(-\hat{r}^o) + \cdots + (-\hat{r}^o)}_{(n-j)/2 \text{ terms}} + \hat{r}^o \\
&= \hat{r}^o.
\end{aligned} \tag{7}$$

Similarly, for an unmatched thought $t_j^{\text{unmatched}}$, the advantage is:

$$\begin{aligned}
A_{\text{index}(j-1)<k\leq\text{index}(j)}^{\text{unmatched}} &= r_{\text{index}(j)} + r_{\text{index}(j+1)} + \cdots + r_{\text{index}(n)} \\
&= \underbrace{\hat{r}^o + \cdots + \hat{r}^o}_{(n-j+1)/2-1 \text{ terms}} + \underbrace{(-\hat{r}^o) + \cdots + (-\hat{r}^o)}_{(n-j+1)/2 \text{ terms}} + \hat{r}^o \\
&= 0.
\end{aligned} \tag{8}$$

$\square$

## C EXPLANATION FOR ADVANTAGE DESIGN

As stated in the paper, we adopt the process-supervised GRPO framework introduced in DeepSeek-Math Shao et al. (2024), where the advantage estimator is computed using Eq. 2 and 3:

$$\hat{r}_i^{\text{index}(j)} = \frac{r_i^{\text{index}(j)} - \text{mean}(R)}{\text{std}(R)}, \quad \hat{A}_{i,t} = \sum_{\text{index}(j) \geq t} \hat{r}_i^{\text{index}(j)} \tag{9}$$

Although this formulation differs from the classical form $A = Q - V$, the normalized term implicitly serves the role of a baseline by subtracting the group mean reward and normalizing variance, making $\hat{A}_{i,t}$ a variance-reduced relative return. Therefore, this relative return aligns with the conceptual purpose of advantage, namely to measure the return of an action relative to the expected return.

Building on this formulation, TP-GRPO modifies the advantage estimator by only normalizing outcome rewards:

$$\hat{r}_{i,o} = \frac{r_{i,o} - \text{mean}(r_{j,o})}{\text{std}(r_{j,o})} \tag{10}$$

This design is motivated by three considerations:

- For correct solutions, we assign a reward of $+r^c$ to correct thoughts and $-r^c$ to incorrect thoughts. This construction ensures that the cumulative return for correct thoughts becomes $R = r_o$, preserving the optimization target as in the outcome-reward-only setting (as stated in Proposition 1), thereby preventing optimization drift caused by introducing process rewards. Normalizing all rewards would violate Proposition 1 and alter the optimization dynamics.

- Normalizing across all rewards, including process-level rewards, would modify the relative scale between correct and incorrect outcome rewards, potentially weakening the role of outcome rewards as the primary global optimization signal. Instead, we aim to retain the original effect of outcome rewards while augmenting them with process-level guidance.

- Under the advantage computation defined in Eq. 3, the resulting advantage for correct thoughts becomes $A = \hat{r}_o$ (as formally proven in Appendix B), which is independent of process rewards and benefits from variance reduction due to normalization. This aligns with the intended behavior of an advantage estimator as a relative, variance-reduced training signal.

## D RELATED WORKS

### D.1 PROCESS REWARD MODELS

Process Reward Models (PRMs) aim to assess the quality of reasoning trajectories by assigning rewards to each evaluation unit—typically a reasoning step—complementing traditional outcome rewards (Li & Li, 2024; Setlur et al., 2024; Zhang et al., 2025). Depending on the evaluation paradigm, PRMs are typically categorized into discriminative and generative approaches (Zhao et al., 2025). Discriminative PRMs learn to predict step-level correctness scores through supervised training (o1 Open Series, 2024b;a). OpenAI's PRM800K dataset (Lightman et al., 2023), with large-scale human annotations, has significantly advanced this area. Wang et al. (2023b); Luo et al. (2024); Wang et al. (2024c); Lu et al. (2024b;a) explore automatic labeling through simulation. However, discriminative PRMs often suffer from poor generalization, vulnerability to reward hacking, and limited interpretability (Guo et al., 2025) due to their reliance on domain-specific training and direct score prediction. In contrast, Generative PRMs treat evaluation as a generation task (Ankner et al., 2024; Zhao et al., 2025). Rather than predicting rewards directly, they analyze whether each reasoning step is correct and use the generation probability of indicative tokens (e.g., "correct", "yes") as rewards (Zhang et al., 2024b; Mahan et al., 2024). These approaches offer greater interpretability and scalability by leveraging LLMs' intrinsic reasoning capabilities (Wang et al., 2023a). Many works adopt an LLM-as-a-Judge framework (Zheng et al., 2023; Gu et al., 2024), prompting LLMs for evaluation without further training (Zheng et al., 2024), and some others enhance evaluation via further training (Wang et al., 2024b; Khalifa et al., 2025; Zha et al., 2025; Xiong et al., 2025).

This work does not seek to explore how training might enhance the performance gains of GenPRM; nevertheless, we believe our framework can achieve additional evaluation improvements via training.

Instead, it targets fundamental issues in the GenPRM workflow that have been overlooked in prior studies. These include whether evaluation should be grounded in reasoning ability or decomposed around other core capabilities, determining suitable reward granularity, and identifying the proper timing for applying process rewards. We hope our findings stimulate broader discussion within the community on these topics. Therefore, our contribution lies primarily in rethinking and innovating the philosophy of generative evaluation and reward design,

## D.2 PRMs for Optimizing Large Reasoning Models

Prior to the release of DeepSeek R1, process rewards are widely explored as a means to improve the reasoning capabilities of LLMs (Zhang et al., 2024a; Wang et al., 2024a). Depending on the application stage, process rewards are typically employed to guide trajectory pruning during inference (Uesato et al., 2022; Kalra & Tang, 2025; Muennighoff et al., 2025) or to shape optimization during training (Zhang et al., 2024c). In this work, we focus on PRM's role in training. Early approaches such as Rest-MCTS* (Zhang et al., 2024a) train PRMs to rank reasoning trajectories, selecting high-quality traces for the learning process of reasoning. (Chen et al., 2024) estimates step rewards via question decomposition, but ultimately integrate them into the outcome reward, limiting the benefit of fine-grained supervision. A similar issue arises in (Li et al., 2024b), where process rewards are not effectively utilized. While OpenR (Wang et al., 2024a) employs PRMs to guide policy iteration in LLM training, it remains confined to the discriminative PRM paradigm. After the release of DeepSeek R1 (Guo et al., 2025), research attention on PRMs shifted toward Generative PRMs (o1 Open Series, 2024a). For instance, ReasonFlux-PRM (Zou et al., 2025) estimates process rewards to guide GRPO optimization but still integrates process rewards to outcome ones without fully utilizing the process rewards. To unlock the full potential of PRMs, several key challenges must be addressed, including ambiguous step segmentation, reward hacking, and continual retraining requirements. In response, we propose a suite of targeted improvements: a comprehension-driven evaluation strategy, a thought-level granularity for assessment, and a capacity-adaptive reward mechanism.

# E  More Experiment Settings

## E.1  Training Settings

For process-level evaluation, we employ Qwen3-32B as the primary evaluator model. To mitigate the substantial computational cost associated with this step, we implement several heuristic rules to filter out solutions that are unlikely to benefit from fine-grained process rewards:

- Solutions that yield correct final answers but have concise reasoning chains (fewer than 4,096 characters) are excluded, as their intermediate steps are presumptively correct.

- Solutions that produce incorrect and very short answers (fewer than 256 characters) are also filtered out, as they seldom contain discernible reasoning steps.

- Solutions truncated due to exceeding maximum length constraints are similarly discarded.

Due to the application of the aforementioned filtering rules and the fact that some solutions receive no process reward after evaluation, the number of effective training samples in each TP-GRPO iteration is often insufficient to train 50 steps (more details can be found in Appendix G). As a result, TP-GRPO performs fewer than 50 steps per iteration and does not save checkpoints at fixed 50-step intervals.

Our implementation of TP-GRPO, as well as all replicated baseline experiments, is built upon the GRPO algorithm. We incorporate two key modifications proposed in DAPO: the $\epsilon_{high}$ parameter and the use of token-level loss. The outcome reward function combines accuracy and format rewards. We extract the answer from "\boxed{}" and compare it against the golden answer: a perfect match yields an accuracy reward of 1, otherwise 0. Additionally, we observed collapse when reproducing the on-policy GRPO baseline with $\beta = 0$. Therefore, following the open-r1 implementation (Hugging Face, 2025), we set $\beta = 0.04$ for the GRPO baseline. Additional hyperparameters and configurations are summarized in Table 6.

Table 6: All key configuration hyperparameters used in this paper.

| Hyperparameter | Value |
|---|---|
| *Sampling Configuration* | |
| system prompt | {problem} Let's think step by step and output the final answer within \boxed{}. |
| temperature | 1.0 |
| max prompt length | 1024 |
| max response length | 8192 |
| rollout size | 8 |
| *Process Evaluation Configuration* | |
| default model | Qwen3-32B |
| backend | VLLM |
| temperature | 0.7 |
| max tokens | 8192 |
| tensor-parallel-size(VLLM) | 2 |
| max_num_seqs(VLLM) | 512 |
| *Training Configuration* | |
| gradient accumulation steps | 1 |
| learning rate | 1.0e-06 |
| lr scheduler type | constant |
| max steps | 50 |
| max iteration num | 20 |
| per device train rollout batch size | 5 |
| process reward coefficient $\alpha$ | 1 |
| $\epsilon_{low}$ | 0.2 |
| $\epsilon_{high}$ | 0.28 |
| *Evaluation Configuration* | |
| max tokens | 32768 |
| temperature | 0.6 |
| top P | 0.95 |
| top K | -1 |
| gpu_memory_utilization | 1.0 |

### E.2 EVALUATION SETTINGS

All evaluation experiments—including those for TP-GRPO and all baselines—are conducted using the evaluation codes from the **rllm** project (Luo et al., 2025).

## F PROCESS EVALUATION DETAILS

In this section, we provide the implementation details of the process evaluation introduced in Section 3.1.

**Overall of our generative process evaluation Pipeline** We first revisit the overall process evaluation pipeline proposed in Section 3.1. As illustrated in Figure 7, the pipeline evaluates solutions in three sequential steps. In the first step, we decompose the reasoning process into individual steps and apply different strategies to determine each step's type based on the correctness of thinking content. In the second step, consecutive correct or incorrect steps are grouped together to form thought-level segments. The rationale behind this approach is supported by the results shown in Section 4.3.1. In the third step, we adaptively assign different rewards to correct and incorrect thought segments based on the model's capability for the given problem, thereby effectively balancing exploitation and exploration during training.

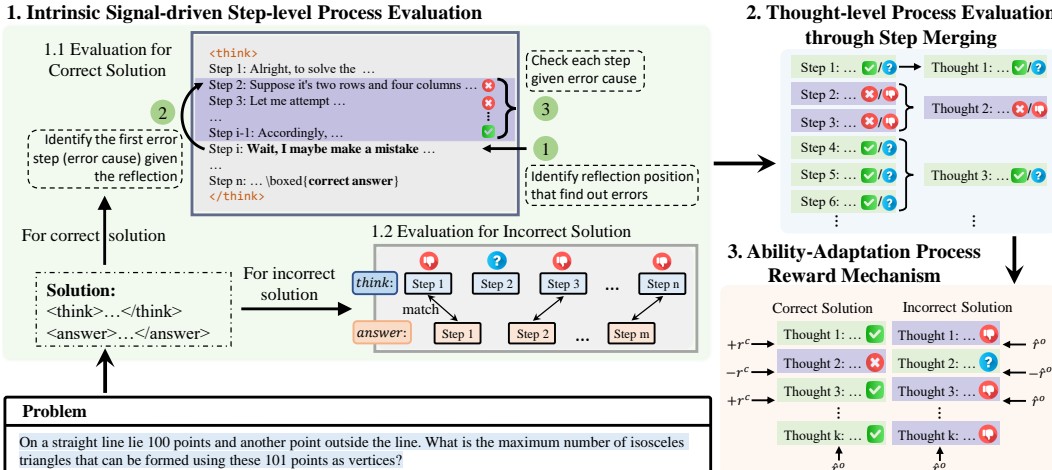

Figure 7: The pipeline of intrinsic signal-driven thought-level generative process evaluation. The whole process consists of three steps: (1) Intrinsic Signal-driven Step-level Process Evaluation, which evaluates each reasoning step using intrinsic signals from solutions; (2) Thought-level Process Evaluation through Step Merging, which merges continuous steps with the same correctness to enable thought-level evaluation; and (3) Ability-Adaptation Process Reward Mechanism, which calculates rewards based on model capability, balancing exploration and exploitation.

**Atomic Logical Actions-Based Decompose** Each solution should first be decomposed into a series of steps with appropriate granularity for subsequent processing. Given that model-generated solutions are often lengthy and unstructured, directly segmenting them into content-complete steps with clear boundaries is challenging. Our decomposition begins by splitting the solution into shorter sentences using the "\n\n" delimiter. However, we observed in practice that the model often generates "\n\n" in the middle of a process, leading to excessive fragmentation. To address this, we curated a list of punctuation marks (detailed in Table 15) that reliably signify the end of a semantically complete, fine-grained sentence. We then sequentially merge consecutive short sentences until the merged segment terminates with one of these specified punctuation marks. For ease of subsequent processing, we then insert a sequentially increasing label (e.g., «0», «1», ...) at the beginning of each merged sentence. To ensure that each sentence has an appropriate semantic granularity, we define eight atomic logical actions and prompt GenPRM to further merge consecutive sentences based on these actions. Ultimately, for each solution we obtain a series of steps with appropriate quantity and granularity, as well as clear semantic boundaries. The prompt used is shown in Table 17. For solutions marked as incorrect, the corresponding final answer also requires decomposition. As answers are considerably shorter than the full reasoning trajectory, we bypass the initial punctuation-based merging and directly apply the atomic logical action decomposition. Specifically, GenPRM is prompted to enclose each identified step within "<step>" and "</step>" tags. It facilitates the subsequent parsing of steps using regular expressions, and mitigates the risk of GenPRM corrupting the original answer (e.g., by rewriting or omitting parts). The prompt for answer decomposition is shown in Table 18.

**Match-Based Step Type Check for Incorrect Solutions** Following the decomposition, each step within an incorrect solution is assessed by performing a semantic match against the corresponding answer (Section 3.1.1). For solutions that remain excessively long (i.e., exceeding 8192 characters), we apply an additional step-level summarization using GenPRM. The prompt used is shown in Table 19. This makes them more concise for subsequent processing. We focus solely on semantic and content similarity, disregarding whether the solution steps themselves are correct or not. The prompt used can be found in Table 20.

**Reflection-Based Step Type Check for Correct Solutions** For solutions that yield a correct final answer, we assess the correctness of each intermediate step using a three-stage methodology: **Localization**, **Traceback**, and **Verification** (Section 3.1.1). First, in the Localization stage, the step-decomposed solution is provided to GenPRM, which is tasked with identifying all reflection segments. To filter out generic validation statements (e.g., "double-checking"), we constrain the

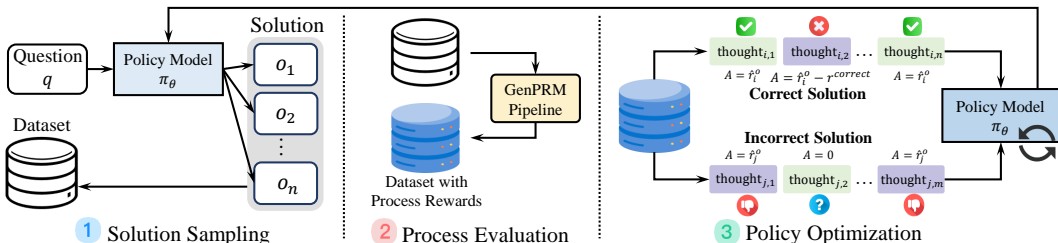

Figure 8: The overview of our proposed efficient off-policy training pipeline. This design ensures full utilization of GPU resources at each stage—all GPUs are either concurrently used for sampling, process evaluation, or parameter updates, avoiding any idle computation.

model to identify only those reflections that pinpoint concrete errors. The prompt for this task is detailed in Table 21. Second, duringTraceback, we iterate through each identified reflection to trace its corresponding error source, defined as the earliest step where the specified error first occurred (see prompt in Table 22). After identifying the source for each reflection, we retain only the one with the earliest index, as this represents the ultimate origin of the error. If multiple reflections map to this same earliest source, we select the reflection with the latest index to define the end of the error-reflection span. All steps within this span are considered semantically related to the source error. Finally, in the Verification stage, GenPRM classifies each step within the identified span into one of three types: **Incorrect**, **Reflection**, and **Correct**.

- **Incorrect:** A step containing a logical or computational error.
- **Reflection:** A step containing a reflective or self-correcting action.
- **Correct:** A step that is neither Incorrect nor a Reflection.

The prompt used for this classification task is shown in Table 23. This multi-stage workflow is implemented by parsing the **JSON** output from each stage and using it as input for the subsequent stage.

## G  OFF-POLICY TRAINING PIPELINE DETAILS

Conventional on-policy RL training alternates between sampling rollouts and performing policy updates, often on separate sets of GPUs. This approach can lead to significant GPU idling and low overall utilization, as policy update hardware waits for the sampling phase to complete. To overcome this inefficiency, we propose an off-policy training framework, depicted in Figure 8, which decouples these phases into three distinct steps:

- **Solution Sampling:** The policy model is deployed using the vLLM backend. A batch of prompts sufficient for **n** update steps (**n**=50 in our experiments) is prepared, and the model generates responses for all prompts. The resulting data is saved locally.
- **Process Evaluation:** The policy model is unloaded from the GPUs, and our GenPRM is deployed, also using the vLLM backend. The process evaluation pipeline, as described in Section 3.1, is then applied to generate process rewards for the data collected in the previous step.
- **Policy Optimization:** Finally, GenPRM is unloaded, and the policy model is reloaded for training. The model is then trained on the offline-collected, reward-annotated data for the target number of steps. This completes one full iteration of our framework, and the cycle is repeated until training converges.

## H  MORE EXPERIMENTS RESULTS

### H.1  HYPERPARAMETERS ANALYSIS

We further analyze the hyperparameter $\alpha$ to assess its impact on model performance. Experiments are conducted on the 1.5B model with $\alpha \in \{0, 0.5, 1, 2, 4\}$, and the results on AIME 2024, AIME

Table 7: Hyperparameter Analysis with different $\alpha$ settings.

| Model | AIME 24 | AIME 25 | AMC 23 | Avg. |
|---|---|---|---|---|
| Base Model | 28.80 | 22.50 | 62.90 | 38.07 |
| TP-GRPO ($\alpha = 0$) | 30.00 | 25.00 | 62.50 | 39.17 |
| TP-GRPO ($\alpha = 0.5$) | 31.46 | 24.17 | 63.78 | 39.80 |
| TP-GRPO ($\alpha = 1$) | 33.12 | 25.63 | 64.01 | 40.92 |
| TP-GRPO ($\alpha = 2$) | 32.29 | 25.00 | 64.46 | 40.58 |
| TP-GRPO ($\alpha = 4$) | 31.46 | 23.33 | 64.38 | 39.72 |

2025, and AMC 2023 are summarized in Table 7. The results show that $\alpha = 1$ achieves the best performance. This can be explained by the advantage computation for incorrect thoughts:

$$A = \hat{r}^o - r^c = \hat{r}^o - \alpha \cdot \text{acc}_G. \tag{11}$$

When $\alpha = 1$, we have $0 < A < \hat{r}^o$, meaning that incorrect thoughts receive positive but reduced advantage values—effectively suppressing their optimization strength relative to correct ones. As $\alpha$ decreases, this regularization weakens; when $\alpha = 0$, the mechanism reduces to outcome-only off-policy GRPO, losing the benefits of process-aware credit assignment for correct solutions. Conversely, when $\alpha > 1$, the penalty intensifies, potentially driving $A < 0$, which may over-suppress erroneous reasoning paths and hinder the LRM's ability to engage in productive trial-and-error exploration.

## H.2 ANALYSIS ON OFF-POLICY TRAINING PIPELINE

In this work, we propose an off-policy training pipeline to reduce training time by reducing GPU idling. According to the results in Figure 3, it is evident that off-policy GRPO underperforms on-policy GRPO in both accuracy and training efficiency, indicating that on-policy training is more advantageous—consistent with conventional training experience. Figure 3 presents a clearer comparison, where on-policy GRPO demonstrates a more pronounced improvement trend over training steps. To further investigate the impact of update frequency in off-policy training, we reduce the update interval of TP-GRPO's off-policy pipeline from 50 steps to 25 steps (and to 1 step in an extreme case, which essentially corresponds to true on-policy updates). We observe that both AIME 24 and AMC 23 tasks show further performance improvements. In summary, although off-policy training is known to be less effective than on-policy training, TP-GRPO trained under the off-policy setting still outperforms the on-policy GRPO baseline. This further validates the effectiveness of our proposed GenPRM mechanism.

Table 8: Analysis study on the impact of off-policy training.

| Model | AIME 24 | AIME 25 | AMC 23 | Training Steps |
|---|---|---|---|---|
| On-policy GRPO | 32.71 | 24.58 | 64.53 | 850 |
| Off-policy GRPO | 30.83 | 24.79 | 62.05 | 950 |
| TP-GRPO (25 steps) | 33.54 | 23.96 | 64.53 | 207 |
| TP-GRPO (50 steps) | 33.12 | 25.63 | 64.01 | 140 |

## H.3 TRAINING EFFICIENCY COMPARISON

We evaluated the training efficiency of TP-GRPO against other high-performing methods, with corresponding hyperparameters summarized in Table 10. Our method demonstrates substantial improvements in training efficiency on both the DeepSeek-R1-Distill-Qwen-1.5B and DeepSeek-R1-Distill-Qwen-7B models. Relative to the baseline GRPO algorithm, TP-GRPO accelerates training by a factor of 4 to 8 while also achieving superior performance, as detailed in Section 4.2.

## H.4 CASE STUDIES

To intuitively compare the improvements of our proposed TP-GRPO algorithm over the original GRPO, we selected one problem each from the AIME24 and AIME25. These problems were then

Table 9: Analysis study on the impact of off-policy training.

| Model | Traing Steps | Batch Size | Rollout Size | *Effic.* |
|---|---|---|---|---|
| DeepSeek-R1-Distill-Qwen-1.5B | | | | |
| AutoThink (Stage 1) | 220 | 128 | 16 | -0.53 |
| AdaptThink | 300 | 128 | 16 | 0.53 |
| Open-RS1 | 100 | 96 | 6 | 0.57 |
| Still-3-1.5B-Previous | 250 | 128 | 8 | 0.98 |
| AReaL-1.5B-Preview(Stage 1) | 200 | 1024 | 8 | 0.13 |
| DS-1.5B-Preview | 1750 | 128,128,128 | 8,16,16 | <0.49 |
| GRPO Replication | 850 | 5 | 8 | 4.65 |
| TP-GRPO | 140 | 5 | 8 | 36.43 |
| DeepSeek-R1-Distill-Qwen-7B | | | | |
| AutoThink (Stage 1) | 220 | 128 | 16 | -0.53 |
| ReasonFlux-F1 | 3750 | 16 | 1 | -1.15 |
| AdaptThink | 150 | 128 | 16 | 0.10 |
| AReaL-boba-RL-7B | 250 | 512 | 16 | 0.37 |
| Skywork-OR1 | 1320 | 256,160 | 16,32 | <0.26 |
| GRPO Replication | 400 | 5 | 8 | 9.6 |
| TP-GRPO | 214 | 5 | 8 | 40.07 |

Table 10: Training hyperparameters for efficiency comparison.

presented to the DeepSeek-R1-Distill-Qwen-1.5B and DeepSeek-R1-Distill-Qwen-7B models, which were trained using both algorithms respectively, to observe their corresponding reasoning processes. The detailed results for the DeepSeek-R1-Distill-Qwen-1.5B model are shown in Table 11 and Table 12. , while the detailed results for the DeepSeek-R1-Distill-Qwen-7B model are shown in Table 13 and Table 14. As observed from the model's thinking content, the model post-trained with the TP-GRPO algorithm can correct the specific reasoning errors made by the model trained with the original GRPO. This provides an intuitive demonstration of the effectiveness of our proposed algorithm.

---

**Case Study for 1.5B Model with GRPO**

**[Question]**
Alice chooses a set $A$ of positive integers. Then Bob lists all finite nonempty sets $B$ of positive integers with the property that the maximum element of $B$ belongs to $A$. Bob's list has 2024 sets. Find the sum of the elements of A. Let's think step by step and output the final answer within \boxed{}.

**[Model Thinking Content]**
Okay, so Alice picks a set A of positive integers. Then Bob lists all finite nonempty sets B of positive integers where the maximum element of B is in A. The total number of such sets B is 2024. We need to find the sum of the elements of A.
...

So, if A has n elements, then the total number of such sets B is the sum from k=1 to n of $2^{k-1}$. Because for each element $a_k$ in A, the number of subsets where $a_k$ is the maximum is $2^{k-1}$. Therefore, the total number of sets B is $2^n - 1$, since the sum from k=0 to n-1 of $2^k$ is $2^n - 1$ So, if A has elements $a_1, a_2, ..., a_n$, then the total number is $2^{a_1-1} + 2^{a_2-1} + ... + 2^{a_n-1} = 2024$
...
Now, 2024 can be expressed as a sum of powers of 2. Let's see, 2024 in binary is 11111101100, which is $1024 + 512 + 256 + 128 + 64 + 32 + 4$. Wait, but let me compute that.
Wait, 1024 is $2^{10}$, 512 is $2^9$, 256 is $2^8$, 128 is $2^7$, 64 is $2^6$, 32 is $2^5$, 16 is $2^4$, 8 is $2^3$, 4 is $2^2$, 2 is $2^1$, 1 is $2^0$. So, $1024 + 512$ is 1536, plus 256 is 1792, plus 128 is 1920, plus 64 is 1984,

plus 32 is 2016, plus 4 is 2020. Wait, that's $1024 + 512 + 256 + 128 + 64 + 32 + 4 = 2020$. Hmm, but 2024 is 4 more, so 2020 + 4 is 2024. So, in terms of powers of 2, $2024 = 2^{10} + 2^9 + 2^8 + 2^7 + 2^6 + 2^5 + 2^2$.

...

So, the exponents $a_i - 1$ are 2, 5, 6, 7, 8, 9, 10. Therefore, $a_i$ are 3,6,7,8,9,10,11. Therefore, the set A is 3,6,7,8,9,10,11. Therefore, the sum of the elements of A is $3 + 6 + 7 + 8 + 9 + 10 + 11$.

...

Therefore, the sum of the elements of A is $3 + 6 + 7 + 8 + 9 + 10 + 11$.
Let me compute that:
3 + 6 is 9.
9 +7 is 16.
 16 +8 is 24.
24 +9 is 33.
33 +10 is 43.
43 +11 is 54.
So, the sum is 54.
Therefore, the answer is 54.
**Final Answer**: The sum of the elements of A is $\boxed{54}$.

Table 11: Case study of DeepSeek-R1-Distill-Qwen-1.5B with GRPO. For the given problem, the model incorrectly answered 54, while the correct answer is 55. Due to the model's extensive thinking process, we have excerpted the key steps for clarity. As shown, the error (highlighted by orange) stems from a miscalculation in the binary representation of 2024.

### Case Study for 1.5B Model with TP-GRPO

**[Question]**
Alice chooses a set $A$ of positive integers. Then Bob lists all finite nonempty sets $B$ of positive integers with the property that the maximum element of $B$ belongs to $A$. Bob's list has 2024 sets. Find the sum of the elements of A. Let's think step by step and output the final answer within \boxed{}.

**[Model Thinking Content]**
Okay, so I have this problem here where Alice chooses a set A of positive integers. Then Bob lists all finite nonempty sets B of positive integers with the property that the maximum element of B belongs to A. Bob's list has 2024 sets. I need to find the sum of the elements of A. Hmm, let me try to break this down step by step.

...

So for each a in A, the number of such sets is equal to the number of subsets of 1, 2, ..., a-1. Because each subset can be combined with a to form a set B with maximum a.
Therefore, for each a in A, the number of sets B with maximum a is $2^{a-1}$. Because the number of subsets of a set with n elements is $2^n$. So in this case, n is a-1, so it's $2^{a-1}$.
So, if A is a set of numbers, say $A = a_1, a_2, ..., a_k$, then the total number of sets B that Bob can list is the sum over each $a_i$ in A of $2^{a_i-1}$. Because each $a_i$ contributes $2^{a_i-1}$ sets. Therefore, the total number of sets B is the sum of $2^{a_i-1}$ for all $a_i$ in A. So, let me denote that $S = \sum_a^A 2^{a-1} = 2024$.

...

So, let's factor 2024 into its prime factors to see if we can represent it as a sum of powers of 2.
First, 2024 divided by 2 is 1012, divided by 2 again is 506, divided by 2 again is 253. 253 is not divisible by 2, so the prime factors are $2^3 \times 253$.
Now, 253 divided by 11 is 23, because 11*23 is 253. So, $2024 = 2^3 \times 11 \times 23$. So, $2024 = 8 \times 11 \times 23$. Hmm, interesting.

But how does this help me? I need to express 2024 as the sum of $2^{a-1}$ terms, each of which is a power of 2, and each a is a positive integer. So, each term is a power of 2, and the sum is 2024. So, to find A, I need to express 2024 as the sum of distinct powers of 2

...

So, let me compute the binary representation of 2024. Calculating:
$2^{10} = 1024$
$2^9 = 512$
$2^8 = 256$
$2^7 = 128$
$2^6 = 64$
$2^5 = 32$
$2^3 = 8$
So, summing these up:
$1024 + 512 = 1536$
$1536 + 256 = 1792$
$1792 + 128 = 1920$
$1920 + 64 = 1984$
$1984 + 32 = 2016$
$2016 + 8 = 2024$
Yes, that adds up correctly. So, 2024 in binary is 11111101000. So, the binary representation has 1s at positions 10,9,8,7,6,5,3. So, that would mean that the exponents in the sum are 10,9,8,7,6,5,3.

...

So, if the exponents are 10,9,8,7,6,5,3, then the corresponding a's are 11,10,9,8,7,6,4.
Therefore, set A is 4,6,7,8,9,10,11.
So, the sum of elements in A is $4 + 6 + 7 + 8 + 9 + 10 + 11$.
Let me compute that:
$4 + 6 = 10$
$10 + 7 = 17$
$17 + 8 = 25$
$25 + 9 = 34$
$34 + 10 = 44$
$44 + 11 = 55$
So, the sum is 55.

...

**Final Answer**: The sum of the elements of A is $\boxed{55}$.

Table 12: Case study of DeepSeek-R1-Distill-Qwen-1.5B with TP-GRPO. Using the same problem from Table 11, we also present only the key steps. Post-training with TP-GRPO, the model successfully computes the binary representation of 2024 and yields the correct final answer. We use blue to highlight these steps.

**Case Study for 7B Model with GRPO**

**[Question]**
Let $A$ be the set of positive integer divisors of 2025. Let $B$ be a randomly selected subset of $A$. The probability that $B$ is a nonempty set with the property that the least common multiple of its elements is 2025 is $\frac{m}{n}$, where $m$ and $n$ are relatively prime positive integers. Find $m + n$. Let's think step by step and output the final answer within \boxed{}.

**[Model Thinking Content]**

Okay, so I have this problem here where I need to find the probability that a randomly selected subset B of the set A (which consists of all positive integer divisors of 2025) has the property that the least common multiple (LCM) of its elements is 2025.

...

I know that 2025 divided by 25 is 81, because 25 times 81 is 2025. And 81 is 9 squared, so 2025 is 25 times 81, which is 5 squared times 3 to the fourth power. So, $2025 = 5^2 3^4$. Therefore, the prime factorization of 2025 is $5^2 3^4$.

Now, to find all the positive divisors of 2025, I can use the exponents of its prime factors. The number of divisors is given by multiplying one more than each of the exponents in the prime factorization. So here, for $5^2$, the exponent is 2, so we have $2 + 1 = 3$. For $3^4$, the exponent is 4, so $4 + 1 = 5$. Therefore, the total number of divisors is $3 \times 5 = 15$. So, set A has 15 elements.

...

Next, we need to consider all possible subsets B of A, and find the probability that the LCM of the elements in B is 2025. Since we're dealing with subsets, the total number of subsets is $2^{15}$, because each element can either be included or excluded. However, the problem specifies that B is a nonempty set, so we have to subtract 1 for the empty set. Therefore, the total number of possible nonempty subsets is $2^{15} - 1$.

Wait, actually, let me think. The problem says "a randomly selected subset of A." It doesn't specify whether it's nonempty or not. But then, it goes on to specify that B is a nonempty set with the property that LCM of its elements is 2025. So, perhaps the probability is considering all subsets, including the empty set, but the event we're interested in is only when B is nonempty. Hmm, the wording is a bit ambiguous.

Looking back at the problem: "The probability that B is a nonempty set with the property that the least common multiple of its elements is 2025." So, it's given that B is a nonempty set, so we can consider the total number of subsets as $2^{15} - 1$, since we exclude the empty set. Therefore, the probability is $\frac{(number\ of\ desired\ subsets)}{(2^{15}-1)}$.

...

Given that $2025 = 3^4 5^2$, the LCM of a subset B is 2025 if and only if the subset contains elements that cover all the prime factors with their maximum exponents.

...

By inclusion-exclusion, this number is $U - |A| - |B| + |A \cap B|$.

So, $|A|$ is the number of subsets without 81. Since 81 is one specific element, the number of subsets without 81 is $2^{14}$.

Similarly, $|B|$ is the number of subsets without 25, which is also $2^{14}$.

$|A \cap B|$ is the number of subsets without both 81 and 25, which is $2^{13}$.

Therefore, the number of subsets containing at least one 81 and at least one 25 is:

$2^{15} - 2^{14} - 2^{14} + 2^{13}$.

Let me compute this:

$2^{15} = 32768$

$2^{14} = 16384$

$2^{13} = 8192$

So, 32768 - 16384 - 16384 + 8192.

Compute step by step:

32768 - 16384 = 16384

16384 - 16384 = 0

0 + 8192 = 8192

So, the number of subsets containing at least one 81 and at least one 25 is 8192.

...

Therefore, the probability is $\frac{8191}{(2^{15}-1)} = \frac{8191}{32767}$.

Now, we need to simplify this fraction. Let's see if 8191 and 32767 have any common factors.

First, let me factorize 32767. I know that 32767 is $2^{15} - 1$, which is 32768 - 1 = 32767.

32767 is a Mersenne number, $2^{15} - 1$. Let's see if it's prime.

Well, $2^{15}$ is 32768, so 32767 is 32768 - 1. Let me check its divisibility.

327677 : 74681 = 32767? Let's check: $7 \times 4000 = 28000$, $7 \times 681 = 4767$, so 28000 + 4767 = 32767. Yes, so 32767 = 7 × 4681.

Now, let's factor 4681. Let me check if 4681 is divisible by 7: $7 \times 668 = 4676$, 4681 - 4676 =

5, so no. Next, 11: 4 - 6 + 8 - 1 = 5, which is not divisible by 11. 13: 4681 ÷ 13. 13 × 360 = 4680, so 4681 - 4680 = 1, so no. 17: 17 × 275 = 4675, 4681 - 4675 = 6, not divisible. 19: 19 × 246 = 4674, 4681 - 4674 = 7, no. 23: 23 × 203 = 4669, 4681 - 4669 = 12, no. 29: 29 × 161 = 4669, same as above. 31: 31 × 151 = 4681? Let's compute 31 × 150 = 4650, plus 31 = 4681. Yes! So, 4681 = 31 × 151.
Therefore, 32767 = 7 × 31 × 151.
Now, let's check if 8191 is divisible by any of these primes: 7, 31, 151.
8191 ÷ 7: 7 × 1170 = 8190, so 8191 - 8190 = 1, so remainder 1. Not divisible by 7.
8191 ÷ 31: Let's see, 31 × 264 = 8184, 8191 - 8184 = 7, so remainder 7. Not divisible by 31.
8191 ÷ 151: 151 × 54 = 8154, 8191 - 8154 = 37, which is less than 151, so not divisible.
Therefore, 8191 and 32767 share no common factors besides 1, so the fraction $\frac{8191}{32767}$ is already in its simplest form.
...
Therefore, m = 8191, n = 32767, and m + n = 40958.
I think that's solid. I don't see any mistakes in the reasoning.
**Final Answer**: $\boxed{40958}$

Table 13: Case study of DeepSeek-R1-Distill-Qwen-7B with GRPO. For the given problem, the model incorrectly answered 40958, while the correct answer is 237. Due to the model's extensive thinking process, we have excerpted the key steps for clarity. As shown, the error (highlighted by orange) mainly from a miscalculation in the number of corresponding subsets.

### Case Study for 7B Model with TP-GRPO

**[Question]**
Let $A$ be the set of positive integer divisors of 2025. Let $B$ be a randomly selected subset of $A$. The probability that $B$ is a nonempty set with the property that the least common multiple of its elements is 2025 is $\frac{m}{n}$, where $m$ and $n$ are relatively prime positive integers. Find $m + n$. Let's think step by step and output the final answer within \boxed{}.

**[Model Thinking Content]**
Okay, so I have this problem here where I need to find the probability that a randomly selected subset B of the set A (which consists of all positive integer divisors of 2025) has the property that the least common multiple (LCM) of its elements is 2025.
...
A is the set of all positive integer divisors of 2025. So, let me factorize 2025 to find out its divisors. 2025 is a number I might recognize as a square. Let me check: 45 squared is 2025 because $45 \times 45$ is 2025. So, $2025 = 45^2$. But 45 itself can be broken down further: $45 = 9 \times 5$, and 9 is $3^2$. So, 2025 is $3^4 \times 5^2$. Therefore, the prime factorization of 2025 is $3^4 \times 5^2$.
Now, to find the number of positive divisors, we can use the formula for the number of divisors based on prime factorization. If a number N has the prime factorization $N = p_1^{a_1} * p_2^{a_2} * ... * p_n^{a_n}$, then the number of positive divisors is $(a_1 + 1)(a_2 + 1)...(a_n + 1)$. So, for 2025, which is $3^4 \times 5^2$, the number of divisors is $(4+1) \times (2+1) = 5 \times 3 = 15$. So, set A has 15 elements.
...
Next, I need to consider the power set of A, which is all possible subsets of A, including the empty set and A itself. The number of subsets of a set with n elements is $2^n$. Since A has 15 elements, the power set has $2^{15}$ subsets. Therefore, the total number of possible subsets B is 32768.
Now, the problem is asking for the probability that a randomly selected subset B has an LCM of 2025. So, I need to find how many such subsets exist where the LCM of their elements is 2025, and then divide that by the total number of subsets, which is 32768.
...
Given that 2025 is $3^4 \times 5^2$, the LCM of a subset will be 2025 if and only if in the subset,

there is at least one number with $3^4$ and at least one number with $5^2$. Wait, is that correct? Or is it sufficient that the maximum exponent of 3 in the subset is 4 and the maximum exponent of 5 is 2?

Yes, exactly. So, for the LCM to be 2025, the subset must include numbers such that the highest power of 3 in the subset is $3^4$ and the highest power of 5 is $5^2$. So, in other words, the subset must contain at least one multiple of $3^4$ (which is 81) and at least one multiple of $5^2$ (which is 25).

...

Therefore, the number of subsets with LCM 2025 is equal to the number of subsets that include at least one multiple of 81 and at least one multiple of 25. So, to compute this, we can use the principle of inclusion-exclusion.

First, let's denote:
- Let S be the set A, which has 15 elements.
- Let X be the set of subsets of S that include at least one multiple of 81.
- Let Y be the set of subsets of S that include at least one multiple of 25.
We need to find $|X \cap Y|$, which is the number of subsets that include at least one multiple of 81 and at least one multiple of 25.
By the principle of inclusion-exclusion:
$|X \cap Y| = |X| + |Y| - |X \cup Y|$ So, let's compute each term.
Total subsets: $2^{15}$.
Subsets without any multiple of 81: So, we need to count the number of subsets that do not include 81. 2025 is $3^4 \times 5^2$, so the divisors are of the form $3^a \times 5^b$ where a is 0,1,2,3,4 and b is 0,1,2.
The number of divisors is 15. The number of divisors that are multiples of 81: 81 is $3^4$, so any multiple of 81 must have a=4. So, the number of such divisors is equal to the number of possible b's, which are 0,1,2. So, there are 3 divisors that are multiples of 81: $81 \times 1 = 81, 81 \times 5 = 405, 81 \times 25 = 2025$.
Therefore, the number of elements not divisible by 81 is 15 - 3 = 12. Therefore, the number of subsets without any multiple of 81 is $2^{12}$.
Similarly, subsets without any multiple of 25: 25 is $5^2$. So, the divisors that are multiples of 25 are those with b=2. The number of such divisors is equal to the number of a's, which are 0,1,2,3,4. So, that's 5 divisors: 1, 5, 25, 75, 225, 2025? Wait, no: 1 is $3^0 \times 5^0$, 5 is $3^0 \times 5^1$, 25 is $3^0 \times 5^2$, 75 is $3^1 \times 5^2$, 225 is $3^2 \times 5^2$, 2025 is $3^4 \times 5^2$. Wait, that's 6 divisors. Wait, but 2025 is $3^4 \times 5^2$, so for b=2, a can be 0,1,2,3,4: that's 5 divisors. Wait, $3^0 \times 5^2, 3^1 \times 5^2, ..., 3^4 \times 5^2$: that's 5 divisors. Wait, why did I think 6? Hmm, maybe I confused with 2025's total divisors.
Wait, no, the number of divisors where b=2 is 5 because a can be 0,1,2,3,4. So, 5 divisors.
Therefore, the number of divisors not divisible by 25 is 15 - 5 = 10. Therefore, the number of subsets without any multiple of 25 is $2^{10}$.
Now, the subsets that include neither any multiple of 81 nor any multiple of 25: So, these are subsets that do not contain any of the 3 multiples of 81 and do not contain any of the 5 multiples of 25. So, how many such elements are there?
Total elements:15.
 Number of elements that are multiples of 81 or 25: Let's compute this using inclusion-exclusion. Number of multiples of 81:3, multiples of 25:5, and multiples of both 81 and 25: Well, 81 and 25 are coprime, so their least common multiple is 81 x 25 = 2025. So, only 2025 is a multiple of both 81 and 25. Therefore, the number of elements that are multiples of 81 or 25 is 3 + 5 -1 =7.
Therefore, the number of elements that are neither multiples of 81 nor multiples of 25 is 15 - 7 = 8.
Therefore, the number of subsets that include neither any multiple of 81 nor any multiple of 25 is $2^8$.
Therefore, putting it all together:
$|X \cap Y|$ = Total subsets - subsets without 81 - subsets without 25 + subsets without both 81 and 25
Which is:

$|X \cap Y| = 2^{15} - 2^{12} - 2^{10} + 2^8$
Let me compute these values step by step.
First, $2^{15} = 32768$
$2^{12} = 4096$
$2^{10} = 1024$
$2^8 = 256$
So, substituting:
$|X \cap Y| = 32768 - 4096 - 1024 + 256$
Let me compute this step by step:
First, 32768 - 4096 = 28672
Then, 28672 - 1024 = 27648
Then, 27648 + 256 = 27904
So, $|X \cap Y| = 27904$
Therefore, the number of subsets B where LCM(B) = 2025 is 27904.
...

Therefore, the probability is $\frac{27904}{32768}$.
Now, we need to reduce this fraction to its lowest terms. Let's find the greatest common divisor (GCD) of 27904 and 32768.
First, let's note that both numbers are divisible by 16: 27904 ÷ 16 = 1744, 32768 ÷ 16 = 2048.
So, 1744 and 2048.
Again, both are divisible by 16: 1744 ÷ 16 = 109, 2048 ÷ 16 = 128.
So, 109 and 128. Now, 109 is a prime number (since it's not divisible by 2,3,5,7,11; $\sqrt{109} \approx 10.4$, so primes up to 11 are tested). 128 is $2^7$. Since 109 is prime and doesn't divide 128, the GCD is 1.
Therefore, the reduced fraction is $\frac{1744}{2048}$. Wait, no: wait, 27904 ÷ 16 = 1744, 32768 ÷ 16 = 2048. Then, 1744 ÷ 16 = 109, 2048 ÷ 16 = 128. So, the reduced fraction is $\frac{109}{128}$.
Wait, but 1744 ÷16=109? Wait, 16*109=1744? Let me compute 16*100=1600, 16*9=144, so 1600+144=1744. Yes, correct. Similarly, 16*128=2048, which is correct.
So, 109 and 128 are coprime, so the reduced fraction is $\frac{109}{128}$.
Therefore, the probability is $\frac{109}{128}$, so m=109, n=128.
Thus, m + n=109 + 128 = 237.
Wait, hold on, but let me confirm the calculations once more because 27904 divided by 32768 reduces by 16 twice, leading to $\frac{109}{128}$, which is correct.
...

Therefore, the probability is $\frac{27904}{32768} = \frac{109}{128}$, so m + n = 237.
Thus, I think that is the correct answer.
**Final Answer**: $\boxed{237}$

Table 14: Case study of DeepSeek-R1-Distill-Qwen-7B with TP-GRPO. Using the same problem from Table 13, we also present only the key steps. Post-training with TP-GRPO, the model successfully computes the number of corresponding subsets and yields the correct final answer. We use blue to highlight these steps.

**Punctuation Marks List**

```
[., ..., ...., ....., ......, !, ?]
```

Table 15: The punctuation marks list we use to merge consecutive pre-split sentences.

**Prompt for LLM-as-a-judge**

I will provide a math problem along with a solution. They will be formatted as follows:
[**Math Problem**]
<math_problem >
...(math problem)...
</math_problem >

[**Solution**]
<step_1 >
...(step 1 of solution)...
</step_1 >

...

<step_n >
...(step n of solution)...
</step_n >

Your task is to review each step of the solution in sequence, analyzing, verifying, and critiquing the reasoning in detail. You need to provide the analyses and the conclusion in the following format:

<analysis_1 >
...(analysis of step 1)...
</analysis_1 >

...

<analysis_n >
...(analysis of step n)...
</analysis_n >

<conclusion >
Correct/Incorrect
</conclusion>

* When you analyze each paragraph, you should use proper verification, recalculation, or reflection to indicate whether it is logically and mathematically valid. Please elaborate on the analysis process carefully.

* If an error is detected in any step, you should describe the nature and cause of the error in detail, and suggest how to correct the error or the correct approach. Once a step is found to contain any error, stop further analysis of subsequent steps (as they may depend on the identified error) and directly provide the conclusion of "Incorrect." For instance, given a solution of five steps, if an error is found in the third step, you should reply in the following format:

<analysis_1 >
...(analysis of step 1)...
</analysis_1 >

<analysis_2 >
...(analysis of step 2)...
</analysis_2 >

<analysis_3 >
...(analysis of step 3; since an error is found here, also provide detailed critique and correction

guideline)...
</analysis_3 >

<conclusion >
Incorrect
</conclusion >

Note that the analyses of steps 4 and 5 should be skipped as the step 3 has been found to contain an error.

* Respond with your analyses and conclusion directly

_______________________________________________________________________________

The following is the math problem and the solution for you task:

[Math Problem]
{question}

[Solution]
{decomposed_think_steps}

Table 16: The prompt used for LLM-as-a-judge.

**Prompt for Decomposing Solution**

You need to read the problem-solving process below, which is divided into some steps using line breaks, with each step labeled by a tag "«»" .

**[Pre Split Format Thought]**
{format solution}

Then, you are required to combine consecutive steps into several logically independent sections. Note that each independent section must belong to one of the eight action categories listed below:

(1) Problem Analysis: Decompose the original problem, clarify known conditions and objectives
(2) Subtask Definition: Establish phased goals (e.g., equation construction, variable definition)
(3) Calculation Solving: Perform numerical operations or symbolic derivations
(4) Reflection Check: Verify logical/computational errors
(5) Answer Validation: Confirm correctness of final/intermediate conclusions
(6) Error Correction: Rectify identified issues
(7) Strategy Shift: Switch to alternative problem-solving approach
(8) Final Conclusion: Summarize and present ultimate answer

**Output Specifications**
- Maintain original thought chain order when outputting sections
- Section format: «start »- «end » Concise explanation the reasons for combining these steps together(max 15 words)
- Must satisfy: $start \leq end < next\_start$

**Special Constraints**
- If several consecutive steps involving calculations, enumerations, proofs, or similar processes are working toward the same objective—such as solving the same equation, proposition, or problem—please group them together into a single section without splitting them apart!
- If several consecutive steps are reflections on or verifications for the same issue, combine them into a single section and do not separate them!

**Example output**
«0 » - «2 »These three steps aim to analyze the question
«3 » - «5 »These three steps is computing the same equation
«6 » - «6 »This step aims to draw the final answer

Table 17: The prompt used to decompose preprocessed solutions. We define eight atomic logical actions and use them to guide the step-wise decomposition.

---

**Prompt for Decomposing Answer**

You will be presented with the following content:

**[Math Problem]**
{problem}

**[Answer]**
{answer of the problem}

**\*\*Task Requirements\*\***
Insert "<step>" and "</step>" into the original answer to break down the above answer into several atomic reasoning steps. Each atomic reasoning step belongs to only one of the following actions:

1. **\*\*Problem Analysis\*\***: Analyze the original problem, organize the conditions, and deepen the understanding of the problem.
2. **\*\*Set Subtask Goals\*\***: Clearly define the specific goal to be solved next, such as setting up an equation.
3. **\*\*Calculating or Solving\*\***: Calculate a variable or solve an equation.
4. **\*\*Self-Reflection\*\***: Reflect on potential mistakes and assess whether previous thoughts contain mistakes.
5. **\*\*Verify Correctness\*\***: Validate whether the final answer or intermediate conclusions are correct.
6. **\*\*Error Correction\*\***: Rectify identified mistakes.
7. **\*\*Switch Thinking\*\***: Change to a different thought for solving the problem.
8. **\*\*Arrive at the Final Answer\*\***: Conclude with the final answer.

**\*\*Output Format\*\***
1. The breakdown of steps should not be too coarse; each step should belong to only one type of action.
2. Separate the reasoning steps by inserting "<step>" and "</step>".
3. Besides inserting the "<step>" and "</step>" tags, do not change the original solution.
4. Do not alter the original solution except for inserting the "<step>" and "</step>" tags; keep the original solution complete.
5. You only need to reply with the complete solution, without outputing any additional content.

Table 18: The prompt used to decompose answer. We used the same set of eight atomic logical actions as detailed in Table 17, with only minor modifications to the phrasing.

> **Prompt for Summarizing Solution**
>
> You will be presented with the following two contents:
>
> **[Math Problem]**
> {problem}
>
> **[Solution]**
> {solution in a list format, each element is a step}
>
> This thought process includes $len(solution)$ Steps in total.
> Your task is to summarize the main content of these all Steps in the solution.
> Make sure each summary clearly describes the key points and the intermediate conclusion of the respective step in the thought process.
>
> Finally, your output should be in standard **JSON** format:
> ```json
> {'Thought Step 0': 'Summary of Step 0',
>  'Thought Step 1': 'Summary of Step 1',
>  ...}
> ```

Table 19: The prompt used to summarize each step of a solution. Note that the input solution has already been decomposed into steps in a list format.

---

**Prompt for Matching Solution and Answer**

You will be given the following three sections:

**[Math Problem]**
{problem}
**[Summarized Solution]**
{summarized solution in a list format, each element is a summarized step}
**[Answer]**
{answer of the problem in a list format, each element is a step in answer}

**\*\*Task Requirements\*\***
1. Establish a mapping relationship between all these solution steps and the answer steps.
2. For each answer step, find all solution steps whose summaries match the content of the current solution step summary.
3. Allow one-to-many or many-to-one correspondence.

**\*\*Matching Criteria\*\***
1. The Answer Step should summarize the corresponding Solution Step, or be a refined articulation of it.
2. If the LAST step of the Solution includes the "final answer" from the Answer, it should at least match the LAST step of the Answer.

**\*\*Output Requirements\*\***
1. Analysis Report:
- Explain the matching basis for each solution step in order.
- Only record clear content correspondences.
2. Mapping Table (**JSON** format):

`{"Solution_Step_Number": [Thought_Step_Numbers], ...}`

- Unmatched thought process steps are not included.
3. Example:
```json
{"0": [1, 3],"1": [5]}
```

Table 20: The prompt used to match each answer step with its corresponding solution step(s). Note that both the solution and the answer have already been decomposed into steps in a list format.

**Prompt for Locating Reflection**

**\*\*Instruction:\*\***
You will receive the following two pieces of content:
**[Math Problem]**
{problem}

**[Decomposed Solution]**
{decomposed solution in a list format, each element is a step}

Note that this thought process has been broken down into multiple steps, documenting the complete path from the initial attempt to the final correct solution. It may include attempts with incorrect reasoning, reflections on those attempts, and subsequent corrections.

**\*\*Your Task:\*\***
Carefully read and analyze the provided thought process to identify all segments where 'an error in previous reasoning steps was discovered through reflection or verification.' For each identified segment, complete the following tasks:
1. Record the Step Index: Indicate the step number where this segment occurs.
2. Identify the Fragment: Clearly specify the exact content of these segments.
3. Summarize the Cause: Analyze and summarize the reason why the error occurred.

**\*\*Output Format Requirements:\*\***
Return your results in the following **JSON** format:
```json
{"Reflection 1": {"Step Index": "The step number where this
segment occurs", "Identified Fragment": "The exact content
of the segment where an error in previous reasoning steps
was discovered through reflection or verification",
"Reason": "A summary of the reason why the error occurred"},
...}
```

**\*\*Notes:\*\***
1. The steps you find MUST have identified a previous ERROR through reflection. If this step ONLY involves double-checking a previous calculation to verify its correctness, without finding a specific error, then ignore this step.
2. If there are no qualifying segments in the thought process, return an empty **JSON** object: "{}".

Table 21: The prompt used to find reflection segments within a correct solution. Note that the input solution has already been decomposed into steps in a list format.

**Prompt for Locating Mistake Source**

You will see a math problem and a pre-decomposed thought process for solving this math problem:
[Math Problem]
{problem}

[Decomposed Solution]
{decomposed solution in a list format, each element is a step}

This solution records the complete mental journey from initial attempts to the final correct answer, including trials of incorrect approaches, reflections, and corrections.

The following paragraph explicitly highlights the discovery of a mistake through reflection:
[Identified Fragment]
{reflection identified fragment}

And the reason of the mistake is summarized as following:
[Reason]
{reflection reason}

Your task is to identify the earliest step in the thought process that is responsible for this mistake.

**Task Requirements**
-The step you find MUST itself actually contain logical or calculation errors. You need to verify the step you have identified CAREFULLY.
- Before stating the earliest step you believe is responsible for this mistake, you need to carefully think and provide your detailed analysis.
- Finally, based on your analysis, output the earliest thought step and step number that you believe is responsible for the mistake, and provide your reason.

**Output Format Requirements:**
Return the final results in the following **JSON** format:
```json
{"Earliest Step": "The earliest step content that is
responsible for this mistake", "Step Index": "The step number
of this thought step", "Reason": "The reason you believe
this step is responsible for this mistake"}
```

Table 22: The prompt used to find the error source based on a reflection segment. This prompt utilizes information returned from the prompt in Table 21.

> **Prompt for Reasoning All Mistakes**
>
> You will be presented with the following content:
> - A math problem:
> **[Math Problem]**
> {problem}
> - A decomposed thought process for solving the problem:
> **[Decomposed solution]**
> {decomposed solution in a list format, each element is a step}
>
> This thought process does not directly lead to the correct answer but includes attempts with incorrect reasoning, reflections, and corrections before arriving at the right solution.
>
> - In Step <reflection start step index> of the thought process, a critical mistake occurs:
> **[Earliest Step]**
> {reflection earliest step}
> - This mistake is identified through reflection in Step <reflection step index>:
> **[Identified Fragment]**
> {reflection identified fragment}
>
> **\*\*Task Description\*\***
> - Your task is to identify the types of thinking steps within the range from step <reflection start step index > to step<reflection step index> that meet the following definitions. You need to check each definition in sequence. Once a definition is met, immediately label it with the corresponding step type and stop checking further.
>
>   **- Reflection Steps:** A step that involves reflecting on or verifying the error and its resulting subsequent mistakes.
>
>   **- Incorrect Step:** A step is considered incorrect if: It contains mistakes itself or it continues the analysis based on the error from above steps without introducing a new, correct approach or making a proper correction.
>
> **\*\*Specific Requirements\*\***
> - For each thinking step, carefully analyze and explain your reason before determining its type.
> - The final output format should strictly adhere to standard **JSON**, for example:
> ```json
> {"Step k": {"Analysis": "The specific analysis for Step k
> before determining its type", "Step Type": ("Incorrect" /
> "Reflection" / "Correct")},...}
> ```
> **\*\*Notes\*\***
> - Ensure that your analysis of each step is well-reasoned and avoids subjective assumptions.

Table 23: The prompt used to determine the category of each step within the source-to-reflection range. This prompt utilizes information returned from the prompts in Table 21 and Table 22.

