# OpenReview forum: "Smarter Not Harder: Generative Process Evaluation with Intrinsic-Signal Driving and Ability‑Adaptive Reward Shaping"
_ICLR.cc/2026/Conference — ICLR 2026 Poster_

### Official Review · Reviewer_LpBz · 2025-10-17

**Soundness:** 1
**Presentation:** 2
**Contribution:** 2
**Rating:** 2
**Confidence:** 3

**Summary:**

This paper tries to address key challenges in incorporating generative process reward models (GenPRMs) into RLVR. The identified challenges are: (1) the requirement for a GenPRM with strong reasoning ability; (2) the risk of reward hacking from over-densified rewards; and (3) the potential discouragement of policy exploration. The authors tackle challenge (1) by using semantic matching to locate correct and incorrect steps. They tackle challenge (2) by merging continuous steps with the same correctness into a larger block. They tackle challenge (3) by using an adaptive process reward assignment. By incorporating the process rewards provided by their framework into GRPO, they proposed TP-GRPO. The main results are evaluated on mathematical reasoning benchmarks.

**Strengths:**

- The paper is clearly written and easy to follow.

- By identifying three crucial challenges for GenPRMs, this paper provides meaningful guidance for future research directions in the field.

- The idea of using semantic alignment for process evaluation is intuitively sound, and it could potentially reduce the difficulty of this task for GenPRM.

**Weaknesses:**

**Lack of Evidential Support**

The paper makes several assertions that lack sufficient supporting evidence. For instance:

- At line 236, the paper states that "static rewards can unintentionally suppress exploration." At line 248, the paper states that "when $\text{acc}_G=0$, $r^c$ vanishes, reducing to outcome supervision GRPO and prioritizing exploration." Why static rewards would suppress exploration? And why, when reduced to outcome supervision, is exploration prioritized? Empirical or theoretical evidence is needed to support these justifications.

- At line 220, the paper states that "If process rewards are assigned at the step level ... which could in turn mislead the optimization to incorrect directions." This conclusion requires further justification. For example, if the process reward is assigned correctly at the step level, would it still mislead the optimization direction?

**Strong Assumptions and Limited Application Scenarios**

- At line 203, the "Identify Effective Steps" component depends heavily on the LRM capacity for accurate self-reflection, a capability not universal among all models. How can the proposed method be applied to those LRMs without self-reflection behavior?

- At line 205, the paper states "During reflection, the LRM typically analyzes the causes of the reflected mistake". This assumes that LRM can correctly diagnose the cause of a mistake during reflection. The case of incorrect self-diagnosis is not addressed.

- Even if the LRM has self-reflection and can correctly identify the location of a previous error, the requirement for the GenPRM to identify "steps dependent on... incorrect previous steps" (line 207) might still raise concerns about its need for non-trivial reasoning capabilities.

- At line 211, the hypothesis that "all steps in the answer are erroneous" does not generally hold. As established by previous work [1], incorrect trajectories often contain valid reasoning steps before the first error occurs.

[1] Let's Verify Step by Step, arXiv 2023.

**Incomplete Experimental Setting**

To robustly support the claim that the evaluation scheme has "low reasoning requirements on the PRM," additional settings should be added: using models like Qwen3-32B, Qwen3-4B, and Gemma-3-12B-it as the GenPRMs without applying the proposed "reducing reasoning requirements" methods.

**Questions:**

- At line 248, the paper states that "$\text{acc}_G$ is the accuracy of the G sampled solutions for the same problem." The definition of $\text{acc}_G$ is confusing. Is it the average accuracy across G sampled solutions for the same problem or something else?

- Also at line 248, the paper states that "when $\text{acc}_G=0$, $r^c$ vanishes, reducing to outcome supervision" I am a little bit confused about why  $r^c=0$ would cause it to reduce to outcome supervision.

- Could $\text{acc}_G$ also be applied to the computation of $r^{ic}_i$ to achieve adaptive penalization according to the difficulty of the problem?

- I am confused about the definition of the advantage function in this paper. According to Equation 3 in the paper, it is more like the definition of returns (or cumulative rewards). According to the definition in [1], the advantage function measures whether an action is better or worse than the policy's default (or expected) behavior. I hope the authors will clear this up for me.

[1] High-Dimensional Continuous Control Using Generalized Advantage Estimation, ICLR 2016.

---

> ### Author Response · Authors · 2025-11-21
> **Response for Reviewer LpBz (part 1)**
>
> We sincerely thank you for your valuable suggestions and the time you spent reviewing our work. We have provided detailed responses to your questions and updated the corresponding content in the paper. We hope these clarifications address your concerns. In addition, we have added extra experiments. If any points remain unclear, we would greatly appreciate your further guidance, and we welcome the opportunity for deeper discussion. Below, we present our detailed responses.
>
> **Q1: Why static rewards would suppress exploration?**
>
> **RA1:** Thank you for pointing this out, and we appreciate the opportunity to clarify our statement. Here, the term “static rewards” specifically refers to **dense, correctness-based process rewards**.
>
> In reinforcement learning, it is well-known that dense and static process rewards tend to provide strong local optimization signals, which encourage the agent to refine already-validated high-reward behaviors rather than exploring new reasoning patterns. This often accelerates convergence around the current policy but suppresses exploratory behavior, potentially leading to local optima. For example, in goal-conditioned RL, [1] shows that simple distance-to-goal process rewards may cause agents to stay on sub-optimal trajectories, reducing exploration motivation.
>
> In the context of LRM training, if incorrect reasoning steps are consistently assigned static and relatively strong negative rewards, the model may quickly converge to conservative step-by-step reasoning strategies that prioritize immediate correctness. While this can improve short-term pass@1, it can limit trial-and-error behavior and reduce reasoning-path diversity, thereby weakening exploration capability. Similar concerns are also raised in recent work [2], which suggests that manually defined process rewards may inadvertently encourage shortcut behaviors.
>
> We also acknowledge that not all process rewards suppress exploration: mechanisms based on entropy regularization [3] or curiosity/novelty bonuses [4,7] are known to promote exploration. Our claim applies only to correctness-driven static rewards, which lack exploration-oriented components; hence the phrasing “may inadvertently suppress exploration” is intended to express a risk rather than a universal conclusion.
>
> We have revised the manuscript to better clarify this point.
>
>
>
>
> **Q2: And why, when reduced to outcome supervision, is exploration prioritized?**
>
> **RA2:** First, this statement holds under the condition that the final answer of the sampled responses is correct. When $acc_G = 0$, we have $r^c = \alpha \cdot acc_G = 0$, meaning no correctness-based process reward is provided; thus, the objective degenerates to outcome-supervised GRPO.
>
> Second, the phrase “more exploration” is defined relative to the setting where process rewards are present. In outcome-supervised GRPO, when the final answer is correct, trajectories that contain trial-and-error behaviors are still assigned positive outcome rewards, implicitly encouraging the model to preserve such exploratory behaviors even if the trial step is incorrect. Moreover, due to the sparse nature of outcome rewards, the model must explore more broadly in order to obtain positive returns, a principle widely recognized in Sutton's reinforcement learning book [9] and also supported by related work [5].
>
> In summary, **dense correctness-based process rewards tend to increase exploitation of known successful patterns, whereas sparse outcome rewards rely more heavily on exploration to obtain returns**. Therefore, when $acc_G = 0$, the learning dynamics naturally lean toward more exploratory behavior.

---

> > ### Author Response · Authors · 2025-11-21
> > **Response for Reviewer LpBz (part 2)**
> >
> > **Q3: If the process reward is assigned correctly at the step level, would it still mislead the optimization direction?**
> >
> > **RA3:** Thank you for raising this important question. This concern is aligned with one of the core motivations of our work, and we appreciate the opportunity to clarify it. This claim refers specifically to the case where **process rewards are dense and assigned at the step level**, which differs fundamentally from the scenario where process rewards accurately reflect thought-level correctness or semantic granularity.
> >
> > **(1) Why correct step-level rewards may still mislead optimization?**
> >
> > Even if step-level correctness labels are accurate, dense reward signals may distort the cumulative return structure used in policy gradient optimization.
> >
> > As illustrated in the middle panel of Figure 2(b), a reward of +1 is assigned for each correct step and -1 for each incorrect step. Although the 4th step itself is correct, the cumulative return becomes negative due to the following five consecutive incorrect steps. This creates situations: $\text{Correct step} \Rightarrow R_t < 0 \quad \text{and} \quad \text{Incorrect step} \Rightarrow R_t > 0$,
> > which is the opposite of the intended optimization target. This misalignment results not from reward mislabeling, but from reward cumulation in dense reward settings, especially when the reward magnitude or sign changes across steps.
> >
> > Thus, even correct step-level rewards do not guarantee that policy gradient updates reinforce correct steps and suppress incorrect ones.
> >
> > **(2) Our proposed mitigation and empirical justification**
> >
> > To resolve this, we propose **Step-Merging**, which merges step-level labels into **thought-level** units, significantly reducing reward oscillation and aligning incentive granularity with reasoning semantics.
> >
> > Empirically, in Figure 5 we compute the mutual information (MI) between each token’s advantage value and its correctness label. As theoretically expected, a rational reward design should produce high MI, indicating consistent optimization direction. However, we observe:
> >
> > - **Low MI** w/o Step-Merging, which means correctness and advantage is misalignment
> > - **Significantly higher MI** with Step-Merging, which means optimization direction keeps consistency with step correctness.
> >
> > This confirms that dense step-level supervision harms optimization, even when reward labels are correct, whereas our thought-level reward design provides a more consistency training signal.
> >
> > **Takeaway: Even accurate step-level rewards can mislead optimization when reward is dense and a Long CoT contains steps with heterogeneous correctness labels. Our Step-Merging approach alleviates this issue both theoretically and empirically.**
> >
> >
> >
> >
> > **Q4: How can the proposed method be applied to those LRMs without self-reflection behavior?**
> >
> > **RA4:** Thank you for the insightful question. First, we would like to clarify that we leverage self-reflection signals for process evaluation **only** on solutions that yield correct final answers. When the final answer is correct, there are only two possible scenarios for the reasoning process:
> >
> > - *All steps are correct:* In this case, no explicit process reward is needed, as the final reward alone sufficiently reinforces all reasoning steps.
> > - *Some intermediate steps are incorrect:* In such cases, due to logical consistency, there must exist corresponding reflection steps that identify and correct those erroneous steps. Otherwise, the propagation of errors would make it highly unlikely to arrive at a correct final answer.
> >
> > Therefore, if an LRM lacks self-reflection capability, the second scenario would not occur; only solutions with entirely correct reasoning steps would be observed, rendering process rewards unnecessary.
> >
> > Second, it is now widely accepted that modern LRM models inherently possess self-reflection capabilities. Indeed, self-reflection is increasingly regarded as a foundational capability of advanced reasoning models. Consequently, when referring to LRM (e.g., OpenAI’s o1, DeepSeek-R1, Qwen3, Gemini 3), it is common to assume they possess this capability.

---

> > > ### Author Response · Authors · 2025-11-21
> > > **Response for Reviewer LpBz (part 3)**
> > >
> > > **Q5: The case of incorrect self-diagnosis is not addressed.**
> > >
> > > **RA5:** Thank you for raising this important point. The underlying cause is consistent with above clarification in **RA4** regarding the use of reflection signals. Our assumption is that if an intermediate step is indeed incorrect but the final solution is correct, then there must exist at least one effective reflection step that correctly identifies and corrects the mistake; otherwise, the error would continue to propagate and it would be extremely unlikely for the model to eventually arrive at the correct final answer.
> > >
> > > Therefore, even if the LRM initially fails to detect the error, misdiagnoses the cause, or incorrectly judges a correct step as wrong, such faulty reflections would still need to be corrected by a subsequent and valid reflection step to ensure that the final answer is correct. Therefore, incorrect self-diagnosis does not invalidate the assumption
> > >
> > >
> > >
> > > **Theoretical Contribution and Self-Contained Evaluation Principle**
> > >
> > > Following the responses to Q4 and Q5, we would like to emphasize the theoretical motivation and conceptual value behind our proposed evaluation mechanism. Unlike prior GenPRM-based approaches that rely heavily on the reasoning capability of an external evaluator, our method is built upon the assumption that a correct Long CoT is logically **self-contained**. Specifically, if an incorrect intermediate step exists, there must also exist at least one reflection step within the same solution that identifies and corrects the mistake; otherwise, the final answer would not be correct.
> > >
> > > Under this assumption, the evaluation can be grounded in the **internal logical coherence** of the reasoning trajectory itself, rather than requiring strong external reasoning supervision. We believe this represents an important conceptual step toward **self-improving** or **self-evolving** AI systems, where the reasoning loop becomes internally verifiable and self-correcting [8].
> > >
> > >
> > >
> > > **Q6: The requirement for the GenPRM to identify "steps dependent on... incorrect previous steps" (line 207) might still raise concerns about its need for non-trivial reasoning capabilities.**
> > >
> > > **RA6:** In step 3 of our step evaluation process, the dependency on GenPRM’s reasoning capability is substantially reduced by two strategies:
> > >
> > > - First, at step 3 we already know both the starting step of the incorrect reasoning chain and the cause of the error, meaning that GenPRM is not required to derive the error causality using its own reasoning ability.
> > >
> > > - Second, within the identified error scope, we employ a straightforward heuristic dependency rule:
> > >
> > >   This rule eliminates the need for GenPRM to evaluate the logical correctness of each step from scratch. Instead, it only needs to determine information-flow dependency between steps, which requires considerably weaker capability compared to full reasoning validation.
> > >
> > > Finally, Table 5 shows that replacing GenPRM with variants of different reasoning capabilities leads to only marginal performance variance, further demonstrating that our evaluation method is effectively decoupled from GenPRM’s reasoning ability.
> > >
> > >
> > >
> > >
> > > **Q7: Incorrect trajectories often contain valid reasoning steps before the first error occurs.**
> > >
> > > **RA7:** Thank you for this insightful comment. You are right that the assumption “all steps in the answer are erroneous” is indeed strong. In fact, we explicitly discuss the rationale behind this design choice in Section 3.1.1.
> > >
> > > Our approach represents a deliberate trade-off:
> > >
> > > - On one hand, in the absence of any process-level supervision, a negative outcome reward uniformly penalizes all steps in an incorrect solution. This can lead to over-penalization. However, Long CoTs often include reasonable attempts that are abandoned before reaching a conclusion, reflecting insufficient deliberation, or "underthinking [6]." These steps should not be suppressed.
> > >
> > > - On the other hand, ideally assigning per-step correctness labels would require the GenPRM to possess reasoning capabilities that exceed those of the actor model, which akin to requiring a teacher to be significantly more capable than the student in order to perfectly assess every step of their reasoning. This places an excessive burden on GenPRM. Furthermore, as tasks grow more complex and the actor LRM’s reasoning improves, GenPRM would require continual training, substantially increasing training costs.
> > >
> > > Therefore, rather than aiming to fully correct every error in an incorrect solution, our method seeks to achieve **effective process-level feedback with minimal capability requirements**. While our current strategy may still mistakenly penalize some correct steps, it mitigates the over-penalization inherent in pure outcome-based rewards.

---

> ### Author Response · Authors · 2025-11-21
> **Response for Reviewer LpBz (part 4)**
>
> **Q8: To robustly support the claim that the evaluation scheme has "low reasoning requirements on the PRM," additional settings should be added: using models like Qwen3-32B, Qwen3-4B, and Gemma-3-12B-it as the GenPRMs without applying the proposed "reducing reasoning requirements" methods.**
>
> **RA8:**
> Thank you for your constructive suggestion. In response, we have added GenPRM-based evaluation methods using the LLM-as-a-judge approach on Qwen3-32B, Qwen3-4B, and Gemma-3-12B-it. For LLM-as-a-judge, we followed the prompting and evaluation procedure described in related work [10]. The experimental results are presented below, and for convenience of comparison, we also present TP-GRPO results using different LLMs as GenPRM.
>
> | Model                    | AIME 24 | AIME 25 | AMC 23 | MATH-500 | Avg. |Step   |
> |--------------------------|---------|---------|--------|----------|-------|-------|
> | TP-GRPO(Qwen3-32B)       | 33.12   | 25.63   |   64.01| 83.81    | 51.64 |140    |
> | TP-GRPO(Qwen3-4B)| 33.33   | 24.58  | 63.78    | 83.62 |   51.33 |155    |
> | TP-GRPO(Gemma-3-12B-it)| 32.71   | 23.33   | 64.76  |  83.73   |  51.13   |125   |
>
>
> | Model                    | AIME 24 | AIME 25 | AMC 23 | MATH-500 | Avg. |Step   |
> |--------------------------|---------|---------|--------|----------|-------|------|
> | LLM-as-a-judge(Qwen3-32B)| 30.41   | 24.58   | 63.28  | 82.61    | 50.22 |118    |
> | LLM-as-a-judge(Qwen3-4B)       |  29.79  |  23.54  | 63.10  |  81.94  | 49.59  | 122  |
> | LLM-as-a-judge(Gemma-3-12B-it)| 30.20   | 22.50   | 61.25  |81.28| 48.75 |226   |
>
>
>
> The current results show that the LLM-as-a-judge approach performs worse than TP-GRPO, demonstrating the effectiveness of our proposed GenPRM mechanism. Moreover, LLM-as-a-judge (Gemma-3-12B-it) performs significantly worse than LLM-as-a-judge (Qwen3-32B). Given that Qwen3-32B and Qwen3-4B both possess stronger reasoning capabilities than Gemma-3-12B-it (Qwen3-32B > Qwen3-4B > Gemma-3-12B-it), this gap further reflects the strong dependence of traditional GenPRM methods on the reasoning ability of the evaluator.
>
>
>
> **Q9: The definition of $acc_G$ is confusing.**
>
> **RA9:** We apologize for the ambiguity. The accuracy term refers to the **per-problem sampling accuracy** defined as:$acc_G$ = #{correct solutions among $G$ solutions} / $G$.
> Thus, if all $G$ sampled solutions for the same problem are incorrect, then $\text{acc}_G = 0$; if all of them are correct, then $\text{acc}_G = 1$. We will revise the paper to make this definition explicit.
>
>
>
>
> **Q10: Why $r^c=0$ would cause it to reduce to outcome supervision.**
>
> **RA10:** Please refer to RA2.
>
>
>
>
> **Q11: Could $acc_G$ also be applied to the computation of $r^{ic}$ to achieve adaptive penalization according to the difficulty of the problem?**
>
> **RA11:** Thank you for the insightful suggestion. Our current design introduces an adaptive reward only for correct solutions. In contrast, for incorrect solutions, we prioritized addressing the over-penalization problem that occurs when only outcome-level negative rewards are used. As pointed out in prior work [6], Long CoT frequently performs underthinking, where partially correct but prematurely terminated attempts exist. These steps should not be penalized equally with genuinely erroneous reasoning.
>
> Besides, our evaluation method cannot reliably identify which individual steps are correct within an incorrect solution. Therefore, defining an exploration–exploitation trade-off for such cases becomes ambiguous and may lead to unstable optimization.
>
> We appreciate the suggestion and will consider incorporating adaptive mechanisms if more reliable fine-grained correctness signals for incorrect solutions become available in future work.

---

> ### Author Response · Authors · 2025-11-21
> **Response for Reviewer LpBz (part 5)**
>
> **Q12: The definition of the advantage function in this paper is confusing.**
>
> **RA12:** Thank you for raising this insightful question. We regret that space limitations prevented us from providing a clearer explanation.
>
> As stated in the paper, we adopt the process-supervised GRPO framework introduced in DeepSeekMath, where the advantage estimator is computed using Eq. (2) and (3):
>
> Eq.(2): $\hat{r}_i^{\mathrm{index}(j)} = \frac{r_i^{\mathrm{index}(j)} - \mathrm{mean}(R)}{\mathrm{std}(R)}$
>
> Eq.(3): $\hat{A}\_{i,t}=\sum\_{\mathrm{index}(j)\ge t}\hat{r}\_i^{\mathrm{index}(j)}$
>
>
> Although this formulation differs from the classical form $A = Q - V$, the normalized term implicitly serves the role of a baseline by subtracting the group mean reward and normalizing variance, making $\hat{A}_{i,t}$ a variance-reduced relative return.
>
> Therefore, this relative return aligns with the conceptual purpose of advantage, namely to measure the return of an action relative to the expected return.
> Building on this formulation, TP-GRPO modifies the advantage estimator by only normalizing outcome rewards:
> $\hat{r}\_{i,o} = \frac{r\_{i,o} - \mathrm{mean}({r\_{j,o}})}{\mathrm{std}({r\_{j,o}})}$
> This design is motivated by three considerations:
>
> - For correct solutions, we assign a reward of $+r^c$ to correct thoughts and $-r^c$ to incorrect thoughts. This construction ensures that the cumulative return for correct thoughts becomes $R=r_o$, preserving the optimization target as in the outcome-reward-only setting (as stated in Proposition 1 of our paper), thereby preventing optimization drift caused by introducing process rewards. Normalizing all rewards would violate Proposition 1 and alter the optimization dynamics.
> - Normalizing across all rewards, including all process rewards, would modify the relative scale between correct and incorrect outcome rewards, potentially weakening the role of outcome rewards as the primary global optimization signal. Instead, we aim to retain the original effect of outcome rewards while augmenting them with process-level guidance.
> - Under the advantage computation defined in Eq.(3), the resulting advantage for correct thoughts becomes $A = \hat{r}_o$ (as formally proven in Appendix B), which is independent of process rewards and benefits from variance reduction due to normalization in the outcome rewards. This aligns with the intended behavior of an advantage estimator as a relative, variance-reduced training signal.
>
> Thank you again for your helpful comment; we have now provided a detailed justification for our advantage estimation strategy in the Appendix C.
>
>
>
> These comprise all of our responses, except for some experimental results that are yet to be added. We warmly welcome any further questions or discussions.
>
>
>
>
> [1] Keeping Your Distance: Solving Sparse Reward Tasks Using Self-Balancing Shaped Rewards
>
> [2] LEARNING A DENSE REASONING REWARD MODEL FROM EXPERT DEMONSTRATION VIA INVERSE REINFORCEMENT LEARNING
>
> [3] REASONING WITH EXPLORATION: AN ENTROPY PERSPECTIVE
>
> [4] EXPLORATION BY RANDOM NETWORK DISTILLATION
>
> [5] Exploration in Deep Reinforcement Learning: A Survey
>
> [6] Thoughts Are All Over the Place: On the Underthinking of o1-Like LLMs
>
> [7] Navigate the Unknown: Enhancing LLM Reasoning with Intrinsic Motivation Guided Exploration
>
> [8] Breaking the Reasoning Barrier A Survey on LLM Complex Reasoning through the Lens of Self-Evolution
>
> [9] Reinforcement Learning: An Introduction
>
> [10] Genprm: Scaling test-time compute of process reward models via generative reasoning

---

> > ### Author Response · Authors · 2025-11-25
> > **Kindly Request for Further Feedback**
> >
> > Dear reviewer,
> > We understand you may be very busy, and we truly appreciate your time and consideration. We believe that we have addressed all the questions you raised as thoroughly as possible. If there are any points we may have overlooked or any inconsistencies you would like us to clarify, we would greatly appreciate your guidance. We would be glad to engage in another round of discussion if needed and sincerely look forward to your feedback.

---

> ### Comment · Reviewer_LpBz · 2025-11-28
> **Official Comment by Reviewer LpBz**
>
> I appreciate the authors' detailed responses, which have addressed most of my concerns. I plan to update my score as soon as the system's edit button is available.

---

> > ### Author Response · Authors · 2025-11-28
> >
> > Thank you very much for your thorough and detailed review, which helps us organize and articulate some theoretical points and motivations more clearly. We also sincerely appreciate your thoughtful reconsideration in raising the score. Finally, thank you for the time and effort you have devoted to reviewing our work.

---

### Official Review · Reviewer_N8ji · 2025-10-30

**Soundness:** 3
**Presentation:** 2
**Contribution:** 3
**Rating:** 4
**Confidence:** 4

**Summary:**

This paper proposes a novel reinforcement learning framework for large reasoning models (LRMs) that enhances reasoning efficiency via process reward modeling (PRM). The authors identify key limitations in existing generative process reward models (GenPRMs), notably their dependence on reasoning ability, over-dense reward signals, and susceptibility to reward hacking. To overcome these issues, the paper introduces an Intrinsic-Signal-Driven Evaluation method and a Thought-Level Ability-Adaptive Reward Mechanism, which together form the proposed TP-GRPO algorithm. Specifically, thought-level

Experimental results on the DeepSeek-R1-Distill-Qwen (1.5B and 7B) models demonstrate significant improvements in training efficiency while maintaining comparable performance. Also, the proposed techniques are less sensitive to the GenPRM's reasoning ability, successfully mitigating the reasoning dependency issue.

**Strengths:**

1. Novelty: The techniques of (1) reflection localization for determining the granularity of thoughts and (2) thought-level advantage adjustment are new to me. The paper also introduces several useful methods in Sections 3.1.1 and 3.1.2.

2. Quality: The paper accurately pinpoints three major weaknesses in current GenPRM designs: excessive dependency on reasoning strength, over-dense reward granularity, and reward hacking. The analysis is precise and well-supported by empirical and conceptual insights.

**Weaknesses:**

1. Clarity: It is better to highlight the contribution of the paper with a more concise title, such as "Improving Reasoning Efficiency via Thought-Level Reflective Reward Shaping." The terms "intrinsic-signal" and "ability-adaptive" are somewhat misleading, as the former can be easily confused with the conventional term "intrinsic reward" [1], and the latter could be replaced with more commonly used terms such as "difficulty-aware," given that "ability" sometimes refers to more general categories of skills like coding, math, tool use, etc.

2. Significance, Quality: In Tables 2 and 3, TR-GRPO is compared only with methods based on outcome reward models (ORMs), and additional comparisons with other PRM baselines are needed, such as [2][3][4]. It is also recommended to include comparisons with stronger ORM methods, such as those in [5].

3. Novelty: The thought-level idea is similar to the one in [4]. It is worth discussing the differences between the two works and making an empirical comparison if applicable.

4. Quality: The effectiveness of TR-GRPO would be more evident if the comparisons in Tables 2 and 3 were made fair by using the same budget or by measuring the number of samples needed to achieve the same overall accuracy.

I think the paper provides many useful techniques, but also has obvious issues, as mentioned above. I would be happy to consider raising my score if the aforementioned concerns are well addressed.

### References

[1]: Pathak, Deepak, et al. "Curiosity-driven exploration by self-supervised prediction." International conference on machine learning. PMLR, 2017.

[2]: Zhao, Jian, et al. "Genprm: Scaling test-time compute of process reward models via generative reasoning." arXiv preprint arXiv:2504.00891 (2025).

[3]: Zhang, Hanning, et al. "Entropy-regularized process reward model." TMLR, 2024.

[4]: Xiong, Wei, et al. "Stepwiser: Stepwise generative judges for wiser reasoning." arXiv preprint arXiv:2508.19229 (2025).

[5]: Chen, Minghan, et al. "Seed-grpo: Semantic entropy enhanced grpo for uncertainty-aware policy optimization." arXiv preprint arXiv:2505.12346 (2025).

**Questions:**

* In Figure 3, the accuracy fluctuates significantly and is not monotonically increasing. I am wondering if this is still the case under different random seeds and whether the conclusion still holds.

---

> ### Author Response · Authors · 2025-11-21
> **Response for Reviewer N8ji (part 1)**
>
> We sincerely appreciate your valuable suggestions and the time you dedicated to reviewing our work. We have carefully considered your suggestions regarding terminology, incorporated comparisons and analyses of the methods you mentioned, and added additional experiments. Below, we provide our detailed responses to the shortcomings and issues you identified.
>
> **Q1: The terms "ability-adaptive" and "intrinsic-signal" are not clear enough.**
>
> **RA1:** Thank you very much for your thoughtful and constructive suggestion. We sincerely appreciate your perspective on improving the clarity and focus of the paper title.
>
> We agree that "ability-adaptive" may lead to unintended interpretations related to general skill domains, and we find your proposed terminology "difficulty-aware" clearer and more appropriate. We will carefully check the conference policy regarding title changes, and if allowed, we would be very willing to revise it accordingly. At the very least, we will update the wording in our future arXiv or camera-ready version.
>
> Regarding "intrinsic-signal", we consider that it may partially overlap with the notion of "intrinsic reward" in a broad sense, as our method derives reward signals mainly from internal properties of the Long CoT rather than external supervision. We avoided using "intrinsic reward" directly because, in conventional RL literature, the term is typically associated with curiosity-driven or exploration-oriented reward formulations, which differ from our intended meaning. We will revisit the term and are open to adopting a clearer alternative based on your suggestion and further discussion.
>
>
>
> **Q2: Additional comparisons with other PRM baselines are needed.**
>
> **RA2:** We sincerely appreciate you for this insightful suggestion. In response, we have added two new experiments to strengthen our empirical evaluation:
> (1) A simple LLM-as-a-judge baseline using the prompt template from [1];
> (2) A publicly released **GenPRM-32B** from [1].
>
> We have cited Stepwiser[2] in our Related Work section. However, the paper has not released their GenPRM or training code, which prevents us from conducting a direct and fair comparison.
>
> The comparison results are shown as following:
>
> | Model                    | AIME 24 | AIME 25 | AMC 23 | MATH-500 |Olympic| Avg. |Step   |
> |--------------------------|---------|---------|--------|----------|-------|------|-------|
> | vanilla GRPO       | 32.71   | 24.58   |   64.53 | 82.70    |  43.67 |49.64 | 850    |
> | TP-GRPO(Qwen3-32B)       | 33.12   | 25.63   |   64.01| 83.81    |  43.91|50.10 |140    |
> | LLM-as-a-judge(Qwen3-32B)| 30.41   | 24.58   | 63.28  | 82.61    | 43.35 | 48.85|118    |
> | GenPRM-32B               | 31.45   | 23.13   | 63.44  | 82.53    | 43.35 | 48.78|262    |
>
>
> The experimental results show that TP-GRPO significantly outperforms the two GenPRM-based methods, providing strong evidence for the effectiveness of our approach. Notably, both baseline GenPRM methods perform worse than vanilla GRPO, indicating that process rewards were not effectively leveraged in these methods. This outcome likely stems from the pitfalls we point out in existing GenPRM-based mechanism, further highlighting the necessity of designing an appropriate process reward mechanism. Moreover, GenPRM-32B, as a specially trained PRM, achieves overall performance comparable to that of LLM-as-a-judge (Qwen3-32B), which further demonstrates that even a strong PRM may fail to realize its potential if the process reward mechanism is poorly designed.
>
>
>
> **Q3: It is also recommended to include comparisons with stronger ORM methods.**
>
> **RA3:** We appreciate your kind suggestion. We would like to clarify that the primary focus of this work is not to propose a  state-of-the-art ORM methods. Instead, our central goal is to **explore a novel design principle for PRM** and **address common issues within the existing GenPRM paradigm**.
>
> To this end, we choose the most representative and easy-implemented RLVR algorithm, i.e. GRPO, as our base algorithm. By integrating our proposed evaluation method with GRPO, we aim to demonstrate a specific hypothesis: that the rational incorporation of process rewards can significantly enhance the training efficiency for LRM by improving the exploitation.
>
> Therefore, our contribution and findings are focused on the GenPRM design methodology and are orthogonal to the specific choices of ORM algorithms. We hope this clarification addresses your concern regarding the scope and positioning of our work.

---

> > ### Author Response · Authors · 2025-11-21
> > **Response for Reviewer N8ji (part 2)**
> >
> > **Q4: The thought-level idea is similar to the one in Stepwiser. It is worth discussing the differences between the two works and making an empirical comparison if applicable.**
> >
> > **RA4:** We are grateful for your valuable suggestion and for highlighting the work of Stepwiser [2]. We acknowledge that both our method and Stepwiser aim to address the limitations of overly fine-grained process rewards. This work has been cited in our Related Work section.
> >
> > However, we would like to emphasize that our proposed method features significant and fundamental differences from Stepwiser [2] in both its design philosophy and technical mechanism. These differences are centered on two core aspects:
> >
> > **(1) The Definition of Rewarding Unit**
> >
> > - **Stepwiser (Chunk-Level):** The "chunk" in Stepwiser is essentially an aggregation of steps based on logical structure (a logical block/unit of sub-goal). While this is an improvement over single-sentence steps, the chunk is still a logical "step" in nature. A significant risk remains: in complex Long CoT, the reward density may still be excessively high, leading to the instability commonly associated with dense process rewards. We validate this issue in Section 4.3.1.
> > - **Our Method (Thought-level):** Our defined "thought" is not based on logical structure but aligns with correctness. Our core strategy, Step-Merging, merges consecutive correct steps into a "Correct Thought" and consecutive incorrect steps into an "Incorrect Thought." This mechanism minimizes the density of process rewards while retaining the minimal necessary process signal required to distinguish between correct and incorrect steps.
> >
> > **(2) Decoupling of Evaluation Granularity and Reward Density**
> >
> > - **Stepwiser (Coupled):** Stepwiser employs chunk-level evaluation and directly assigns rewards at the chunk-level. The evaluation granularity and reward density are coupled. This introduces a dilemma: if the chunk size is coarse, correct sub-steps may be unfairly penalized alongside an incorrect one; if the size is too fine, the reward density increases, risking training instability.
> >
> > - **Our Method (Decoupled):** Our approach achieves a crucial innovation: decoupling evaluation granularity from reward assignment density. We perform precise correctness evaluation at the fine-grained Step-level. Subsequently, we use the Step-Merging strategy to assign the reward at the Thought-level. This perfectly resolves the core dilemma in GenPRM: we ensure evaluation accuracy (via fine-grained evaluation) while successfully mitigating the risk associated with dense rewards (via coarse-grained, thought-level reward assignment).
> >
> > In summary, our approach differs fundamentally from Stepwiser in both philosophy and methodology. While Stepwiser is an excellent contribution and achieves meaningful improvements within the traditional GenPRM framework, we argue that it still inherits several inherent limitations of that paradigm. In contrast, we aim to introduce a new generative process evaluation paradigm: on the one hand, enabling step-level evaluation driven by intrinsic signals to reduce dependence on external evaluators; and on the other hand, defining an appropriate reward assignment granularity to mitigate the bias caused by dense rewards. Ultimately, we design a process reward computation mechanism that better balances exploration and exploitation.

---

> > > ### Author Response · Authors · 2025-11-21
> > > **Response for Reviewer N8ji (part 3)**
> > >
> > > **Q5:  The effectiveness of TR-GRPO would be more evident if the comparisons in Tables 2 and 3 were made fair by using the same budget or by measuring the number of samples needed to achieve the same overall accuracy.**
> > >
> > > **RA5:** We deeply appreciate this constructive suggestion, which is very reasonable. We agree that a fair comparison is essential for demonstrating the effectiveness of our approach.
> > >
> > > To rigorously demonstrate the training efficiency of TP-GRPO, our comparison in Table 2 and 3 focuses on two principal aspects:
> > >
> > > (1) Rigorous Comparison with Vanilla GRPO Baselines: Our primary comparison is against the vanilla GRPO algorithm, including both the on-policy and off-policy settings. In these comparisons, all parameters are kept consistent except for the utilization of our proposed process reward mechanism.
> > >
> > > (2) Efficiency Comparison: To comprehensively evaluate training efficiency, we first introduce the metric $Effc.$, which quantifies the magnitude of the performance gain per 100000 training samples. The total number of samples utilized by all compared methods is organized in Table 10 of the Appendix. On the $Effc.$ metric, TP-GRPO (36.43) is significantly higher than vanilla GRPO (4.65).
> > >
> > > For a clearer comparison **by measuring the number of samples needed to achieve the same overall accuracy**, as illustrated in Fig 3, we specifically identified the training step where TP-GRPO first exceeds the maximum performance achieved by the vanilla GRPO. The vanilla GRPO reaches its peak performance at 850 steps, while TP-GRPO surpasses this performance in just 140 steps. This $\frac{850}{140} \approx 6.07$ times reduction in required training steps demonstrates that TP-GRPO achieves superior training efficiency.
> > >
> > > **Q6: In Figure 3, the accuracy fluctuates significantly and is not monotonically increasing. I am wondering if this is still the case under different random seeds and whether the conclusion still holds.**
> > >
> > > **RA6:** We appreciate your careful observation. We acknowledge that results presented in Figure 3 indeed exhibit fluctuation. We attribute this high fluctuation primarily to two implementation constraints imposed by our limited computational resources:
> > >
> > > 1. **Small Batch Size:** Due to hardware constraints, we were forced to utilize a small batch size (5), which is significantly smaller than typical RLVR training setups (e.g., DeepScaler-1.5B-Preview sets as 128).
> > > 2. **Off-Policy Implementation:** The specific implementation of TP-GRPO is off-policy (detailed in Appendix F). Off-policy updates generally introduce greater instability compared to on-policy methods.
> > >
> > > Despite the visible variance, the overall trend of the TP-GRPO curve demonstrates a clear and sustained upward trajectory, with significantly faster convergence and superior final performance compared to the vanilla GRPO baseline.
> > >
> > > To directly address your concern regarding robustness to random seeds, we conducted an additional experiment using a different random seed. The results confirm the robustness of our conclusions.
> > >
> > > | Seed                     | AIME 24 | AIME 25 | AMC 23 | MATH-500 |Olympic| Avg. |Step   |
> > > |--------------------------|---------|---------|--------|----------|-------|------|-------|
> > > | 42(original)             | 33.12   | 25.63   | 64.01  | 83.81    | 43.91 | 50.10|140    |
> > > | 24                       | 33.10   | 25.41   | 64.84  | 83.06   | 43.88 |50.06 | 152   |
> > >
> > >
> > >
> > > The new experimental results further confirm TP-GRPO's effectiveness regardless of the initial random seed.
> > >
> > > [1] Genprm: Scaling test-time compute of process reward models via generative reasoning.
> > >
> > > [2] Stepwiser: Stepwise generative judges for wiser reasoning.

---

> > > > ### Author Response · Authors · 2025-11-25
> > > > **Kindly Invitation for Further Feedback**
> > > >
> > > > Dear Reviewer,
> > > >
> > > > We understand that you are very busy, and we truly appreciate your time. We have carefully reviewed your comments and have made our best effort to address all the points raised in detail. As we found that the rebuttal system does not allow us to modify the title, the change from “ability-adaptive” to “difficulty-aware” will be applied in the future arXiv version.
> > > > If there are any aspects that remain unclear or if you believe additional clarification is needed, we would greatly appreciate your indication. We are fully willing to provide further explanations and would be glad to continue the discussion.
> > > >
> > > > We sincerely look forward to your response and feedback.

---

> ### Comment · Reviewer_N8ji · 2025-11-27
>
> Thanks for the detailed explanation. This addresses my concerns, and I would like to raise my score to 6 (the system edit button seems missing; will do so when it becomes available). It would be nice if the authors could include the aforementioned results and discussions in the later updated revisions.

---

> > ### Author Response · Authors · 2025-11-28
> >
> > We sincerely appreciate your thoughtful suggestions and your decision to raise the score. Thank you for recognizing our detailed explanations. The additional results and discussions you highlighted have already been incorporated into the revised manuscript. We truly appreciate your constructive feedback and the time you have devoted to our paper.

---

### Official Review · Reviewer_iXx8 · 2025-10-31

**Soundness:** 2
**Presentation:** 3
**Contribution:** 3
**Rating:** 6
**Confidence:** 4

**Summary:**

This paper revisits GenPRM for RL of large reasoning models by using intrinsic semantic cues in the solution trajectory to judge step correctness (locating reflections, tracing the earliest error, and interval labeling), merging consecutive steps with the same correctness into “thoughts,” and applying thought-level, ability-adaptive rewards; on incorrect solutions, only thoughts semantically aligned with the wrong answer are penalized. Combined with GRPO as TP-GRPO, the method attains higher or comparable accuracy with far fewer training solutions on DeepSeek-R1-Distill-Qwen 1.5B/7B (e.g., 5.6K vs 34K; 8.56K vs 16K) and introduces the Effic. metric to quantify improved sample efficiency.

**Strengths:**

The research question and proposed method are interesting, e.g., alleviating GenPRM’s reasoning burden and the observation on dense step-wise rewards.

**Weaknesses:**

The method's reliance on the LRM's own capacity for self-reflection and annotation is a fundamental limitation. This approach is inherently self-limiting, as any improvement is capped by the model's existing capabilities and the task's difficulty, hindering the acquisition of novel skills. The poor performance in Table 1 exemplifies this concern.

The experimental setup is questionable. The RL training is remarkably inefficient, yielding only a 2-4% accuracy gain over 400-1000 iterations—a potential artifact of an undersized group size for GRPO that undermines the paper's credibility. Furthermore, the efficiency comparisons in Tables 1 and 2 are inequitable due to mismatched hyperparameters (e.g., group_size, batch_size) across baselines, and a performance comparison at convergence is conspicuously absent.

**Questions:**

N/A

---

> ### Author Response · Authors · 2025-11-21
> **Response for Reviewer iXx8 (part 1)**
>
> We sincerely thank you for recognizing the strengths of our work and for raising a series of insightful questions, as well as the time and effort you devoted to reviewing our paper. We greatly appreciate the opportunity for such an in-depth discussion. Below, we provide our responses to your questions and the issues you highlighted.
>
> **Q1: The method's reliance on the LRM's own capacity for self-reflection and annotation is a fundamental limitation. This approach is inherently self-limiting, as any improvement is capped by the model's existing capabilities and the task's difficulty, hindering the acquisition of novel skills. The poor performance in Table 1 exemplifies this concern.**
>
> **RA1:** Thank you for raising this important question. We appreciate the opportunity to articulate our understanding of intrinsic signals (e.g., reflection cues) in a more systematic manner. We hope that the following discussion helps establish a clearer and more aligned understanding of the motivation and theoretical foundation of our proposed method.
>
> Our response is organized into four parts:
>
> 1. The fundamental limitations of existing GenPRM-based approaches and why they motivate the search for a new process-evaluation mechanism;
> 2. Why our intrinsic-signal–based mechanism is theoretically sound;
> 3. Why our mechanism is *not* self-limiting;
> 4. How results in Table 1 should be interpreted and what the core motivation of this work is.
>
>
>
> **1) Existing GenPRM approaches exhibit a fundamental limitation due to their heavy reliance on reasoning ability**
>
> Current GenPRM-based methods commonly require GenPRM to make evaluations via thinking; that is, to use reasoning ability to assess whether the reasoning step is correct. This mechanism implicitly assumes that GenPRM can correctly solve the original problem or at least its subproblems so that it can detect logical errors in Long CoT. Consequently, these methods rely on a strong assumption: **the reasoning capability of GenPRM should not be weaker than that of the actor model**.
>
> Moreover, because the evaluation process heavily relies on GenPRM’s reasoning capabilities, GenPRM must be continuously upgraded alongside the actor model to remain effective. This creates significant costs in training, maintenance, and scalability.
>
> Based on the above observations, we sought to explore an alternative path: to leverage the existing and robust fundamental capabilities of GenPRM to perform process evaluation as much as possible. To achieve this, we hope to moderately decouple the required evaluative capabilities from the inherent reasoning capabilities. This means we no longer rely heavily on GenPRM's internal reasoning knowledge for the evaluation. To compensate for the information deficit caused by this decoupling, we propose to utilize the key intrinsic signals that are inherently present in the Long CoT.
>
> Furthermore, in the context of the increasingly active research on LLM self-evolution, models are required to evaluate their own generated responses. If evaluation remains heavily dependent on the inherent reasoning capability, this "reasoning evaluating reasoning" can lead to **systemic evaluation bias**. In contrast, leveraging the model's other foundational capabilities to perform the evaluation could theoretically mitigate this bias. From this perspective, our proposed evaluation mechanism, which is based on **semantic** and **reflection signals**, is crucial for achieving **autonomous improvement** without external intervention.
>
>
>
> **2) The intrinsic-signal–based process evaluation mechanism is theoretically well-founded**
>
> For **correct solutions**, we use reflection signals to locate errors. A correct Long CoT generally satisfies two key properties:
>
> - **Consistency between Reasoning Process and Answer:** Math reasoning typically yields specific numerical results, rather than a "lucky guess." The probability of reaching a correct final answer with an incorrect reasoning process is extremely low.
>
> - **Self-Contained Nature of Errors and Reflections:** If an erroneous step exists within a correct solution, it **must** be identified and corrected by a reflection. Otherwise, the error would propagate, leading to an incorrect final answer.
>
> Therefore, we can conclude that the **error steps and corresponding reflection signals appear as a coupled pair** in a correct solution. This crucial insight allows us to reliably locate errors using reflection signals, without having to rely on external reasoning capability.
>
> Furthermore, **the effectiveness of the reflection step is independent of the actor LRM's reflection ability**. If the actor fails to reflect effectively at an intermediate stage, the error will simply propagate and result in an incorrect final answer. Therefore, given the final answer is correct, there must have been at least one effective reflection that successfully identified the corresponding error. This ensures the inherent reliability of our mechanism.

---

> > ### Author Response · Authors · 2025-11-21
> > **Response for Reviewer iXx8 (part 2)**
> >
> > For **incorrect solutions**, our method involves relying on the semantic matching between the "thin" and "answer" segments to perform coarse-grained error localization (Further details are provided in Section 3.1.1.).
> >
> > We acknowledge that this method yields only coarse evaluation. However, we argue that achieving accurate evaluation inherently requires strong reasoning abilities. In contrast, our approach strategically focuses on leveraging only the foundational capabilities of GenPRM, relying predominantly on semantic matching rather than complex reasoning. This design choice significantly reduces the computational burden placed on the GenPRM.
> >
> > **3) The intrinsic-signal–based mechanism is not self-limiting**
> >
> > We would like to clarify that the evaluation quality of our method is fundamentally independent of the actor LRM’s reasoning ability, and instead relies on the intrinsic properties within Long CoT:
> > - For correct solutions, we evaluate steps using the reflection signals in Long CoT. Reflection is widely regarded as a basic capability of LRMs: whenever an incorrect step occurs, Long CoT typically contains valid self-reflection. Therefore, the evaluation does not rely on the overall reasoning capability of the actor LRM.
> >
> > - For incorrect solutions, we conduct evaluation throught matching the “think” and “answer” segments, requiring only that the model output be naturally separable into these two parts, which is an assumption that holds reliably in practice. Thus, the evaluation is also independent of the model’s reasoning strength.
> >
> > In summary, the proposed GenPRM-based evaluation mechanism is conceptually independent of the reasoning capability of the actor LRM.
> >
> > In contrast, existing methods that need the GenPRM’s reasoning capacity introduce an inherent scalability issue: as the actor LRM advances, the GenPRM must maintain a commensurate level of reasoning proficiency for successful evaluation, thereby imposing a systemic constraint.
> >
> > Regarding your apprehension that the actor LRM may be hindered from learning novel skills, we assert that the learning process hinges on two factors: (1) the ability to generate Long CoT with novel thoughts; and (2) high process rewards to reinforce these novel thoughts. Since the reflection signals are strictly correlated with incorrect steps, we think LRM using our evaluation mechanism can still acquire new capabilities.
> >
> > We sincerely appreciate your thoughtful feedback and would be glad to discuss further.
> >
> >
> >
> > **4) Interpretation of Table 1 and the actual focus of our work**
> >
> > We agree with your observation: in Table 1, TP-GRPO do not significantly outperform vanilla GRPO in terms of reasoning accuracy. In fact, the primary contribution of this work is **not** to achieve best converged performance. Instead, our main focus is training efficiency. This work aims to investigate whether process rewards can enhance exploitation and thereby improve the training efficiency of actor LRMs.
> >
> > As Table 1 shows, TP-GRPO surpasses vanilla GRPO (trained for 850 steps) after only 140 steps. This result demonstrates the significant improvement in training efficiency achieved by TP-GRPO.
> >
> >
> >
> >
> >
> > **Q2: The RL training is remarkably inefficient, yielding only a 2-4% accuracy gain over 400-1000 iterations—a potential artifact of an undersized group size for GRPO that undermines the paper's credibility. And a performance comparison at convergence is conspicuously absent.**
> >
> > **RA2:** Thank you for raising this concern. We agree that the training curves in Figure 3 appear to show slow RL improvement, this is primarily due to the tiny batch size used in our experiments under computational constraints.
> >
> > **1) Why RL appears inefficient in our figures**
> >
> > Prior work such as DeepScaler-1.5B-Preview uses a **batch size of 128**, while we were only able to use **batch size = 5** because (i) our computational resources are limited, and (ii) Stage I process evaluation is particularly time-consuming. Under this mismatch, 1000 GRPO training steps in our setting correspond to only $ 1000 \times \frac{5}{128} \approx 40$ optimization steps when batch size=128. Hence, the low efficiency in the curve reflects the influence of batch-size, rather than our deliberate attempt to reduce the effectiveness of GRPO.
> >
> > **2) Evidence that TP-GRPO improves sample efficiency**
> >
> > Even with our very small batch size, TP-GRPO reaches performance **within 140 steps** that surpasses on-policy GRPO trained for **850 steps**. This indicates that the training efficiency of TP-GRPO is approximately $\frac{850}{140} \approx 6.07 \times$ times that of vanilla GRPO. Furthermore, this performance is equivalent to achieving a $+4.32$ score increase on AIME24 with only 6 training steps under batch size=128. This result significantly surpasses the effect reported by DeepScaler-1.5B-Preview. We believe this clearly demonstrates that TP-GRPO significantly improves training efficiency, which is the main focus of our work.

---

> > > ### Author Response · Authors · 2025-11-21
> > > **Response for Reviewer iXx8 (part 3)**
> > >
> > > **3) Why we did not report fully converged curves**
> > >
> > > We acknowledge that our experiments do not reach full convergence. For comparison, DeepScaler-1.5B-Preview shows **no convergence even after 1000 steps with batch size = 128**. Under our batch size = 5 setup, achieving an equivalent amount of training samples would require $1000 \times \frac{128}{5} = 25600$ steps which is far beyond our available compute budget. Thus, fully converged results are infeasible under our device constraints.
> > > As emphasized in the introduction, our goal is to investigate **whether appropriate process rewards can improve exploitation and thereby improve LRM training efficiency**. Demonstrating that process rewards yield higher final converged accuracy would require significantly more compute and is not the intended contribution of this paper. We also hope to conduct further research on this issue in the future, provided that resources permit.
> > >
> > >
> > >
> > >
> > >
> > > **Q3: The efficiency comparisons in Tables 1 and 2 are inequitable due to mismatched hyperparameters (e.g., group_size, batch_size) across baselines.**
> > >
> > > **RA3:** Thank you for pointing out this critical issue. First, we did account for factors such as batch size and group size during computing training efficiency and present the organized results in Table 10 of the appendix. And our reported $Effic.$ metrics are derived from the results summarized in Table 10. Of course, $Effic.$ for works like DeepScaler-1.5B-Preview are used only as references, especially since our implementation differs from these baselines in several algorithmic settings such as the KL coefficient and entropy regularization, making an entirely fair comparison difficult.
> > >
> > > Therefore, **our primary efficiency comparison is performed against the reproduced vanilla GRPO under identical hyperparameters**, where the only difference is whether process rewards are applied. Our comparison consists of two parts:
> > >
> > > 1. We introduce and report the efficiency metric $Effc.$, which measures the performance improvement per 100000 training samples. A higher $Effc.$ indicates better sample efficiency. Under this metric, TP-GRPO significantly outperforms vanilla GRPO.
> > > 2. We further compare learning curves in Figure 3. Vanilla GRPO reaches its best performance at around 850 steps, whereas TP-GRPO achieves the same performance by step 140, providing additional evidence of the efficiency advantage of TP-GRPO.

---

> ### Comment · Reviewer_iXx8 · 2025-11-28
>
> Thank you for the detailed response. After careful consideration, I have the following thoughts:
>
> 1.  **On the GRM's capabilities:** I don't fully agree with the premise that "the GRM must not be weaker than the LRM." This is a classic example of discrimination being simpler than generation. For instance, while solving a complex multiplication like `12309810391 * 1293801823901` is difficult, it is trivial to verify that any result not ending in a '1' is incorrect.
>
> 2.  **On the limits of self-reflection:** I acknowledge the effectiveness of the method our paper in helping the LRM eliminate some errors. However, the core issue remains: your "GRM" is fundamentally dependent on the content that the LRM can explicitly decode and reflect upon. If a theorem, a numerical method, or a line of reasoning is beyond the LRM's current capacity for introspection or has not been explored, it cannot be learned via this mechanism. Its potential is ultimately bounded by the LRM's existing, expressible knowledge.
>
> 3.  **On the efficiency comparison:** I acknowledge the resource constraints you mentioned. However, comparing efficiency directly across experiments with different hyperparameters (like `batch_size` and `rollout_size`) is fraught with risk.  For any given task, we cannot expect performance to double simply by doubling the batch size or rollout size while keeping the step count constant; or halving the steps could even lead to worse results than the original. Therefore, the most convincing evidence comes from the direct, fair comparison against your replicated GRPO baseline under identical settings. (As a side note, given the nature of GRPO, a larger `rollout_size` would have ideally strengthened this baseline comparison, resources permitting).
>
> Taking all these points into consideration, I am inclined to maintain my score of 6, which I consider to be slightly positive.

---

> > ### Author Response · Authors · 2025-11-28
> >
> > We sincerely appreciate your response and your decision to maintain a positive evaluation. Your new comments are highly valuable for further promoting a deeper discussion of the mechanisms of GenPRM.
> >
> > **Q4: On the GRM's capabilities**
> >
> > **R4:** Your concern regarding “the GenPRM must not be weaker than the LRM” is crucial and worthy of in-depth discussion. First, we understand your example: for obvious mistakes (e.g., simple calculation errors), even a relatively weak PRM may correctly identify them. However, we argue that such errors are no longer the primary bottleneck of current LRMs for two reasons:
> >
> > 1. With increasing model capabilities, calculation errors become increasingly rare, while the main deficiencies of LLMs lie in reasoning chains, logical connections, and strategy selection;
> > 2. For arithmetic-type subtasks, external tools can often handle the computation directly.
> >
> > Thus, we believe the core challenge remains improving the correctness of reasoning thoughts.
> >
> >
> >
> > Regarding the common belief that “discrimination is simpler than generation,” we argue that this does not necessarily hold for GenPRM.
> >
> > - The LRM’s reasoning process can be formalized as $p([s_1, \dots, s_k] \mid x)$.
> > - GenPRM’s evaluation of step $s_k$ can be formalized as $p(\text{Correct}, [t_1,\dots,t_m] \mid x, [s_1,\dots,s_k]) =
> >    p(\text{Correct} \mid x,[s_1,\dots,s_k], [t_1,\dots,t_m]) \cdot
> >    p([t_1,\dots,t_m] \mid x,[s_1,\dots,s_k]),$
> >    where $[s_1, s_2, …, s_k]$ is the LRM reasoning steps, and $[t_1,\dots,t_m]$ is GenPRM’s internally generated thought used to judge the correctness of $s_k$.
> >
> > This formulation illustrates two key points:
> >
> > - **GenPRM conducts the discrimination task through generation**.
> > - To correctly evaluate a reasoning step, GenPRM needs to internally know the correct reasoning trajectory and then contrast it with the LRM’s reasoning.
> >
> > Therefore, GenPRM inherently requires reasoning capabilities comparable to or stronger than the LRM. We acknowledge that a weaker GenPRM may still catch errors in some cases, but from the perspective of statistical robustness and evaluation reliability, a weaker GenPRM evaluating a stronger LRM is unstable.
> >
> > Analogously, if a weak-performing student A and a great-performing student B review each other’s examinations, A might occasionally detect B’s mistakes, but B is intuitively more likely to accurately identify A’s errors.
> >
> >
> >
> > **Q5: On the limits of self-reflection**
> >
> > **R5:** You correctly pointed out that GenPRM evaluates LRM’s reasoning based solely on the content already produced by the LRM, and thus cannot introduce new external knowledge. We agree with this point. However, we think this not as a limitation of self-reflection, but as a natural property of GenPRM’s role:
> >
> > - GenPRM acts as a **judge**, not a **teacher**. It provides process-level rewards, not external domain knowledge such as new theorems or formulas;
> > - GenPRM's goal is to help the LRM make better use of the reasoning rollout, not to expand the knowledge frontier.
> >
> > Of course, your suggested extension is indeed a highly interesting idea: using an external model that simultaneously (i) evaluates the LRM’s reasoning as a GenPRM, and (ii) provides additional knowledge as a teacher to help reconstruct improved reasoning trajectories. This hybrid approach may inspire new paradigms in reasoning-oriented reinforcement learning.
> >
> >
> >
> > **Q6: On the efficiency comparison**
> >
> > **R6:** Your concern about fair comparison is very reasonable. And we confirm that our experimental setup strictly follows consistent configurations across methods:
> >
> > - In Tables 1 and 2 and Figure 3, TP-GRPO and vanilla GRPO use the **same batch size (=5)** and the **same rollout size (=8)**.
> > - Figure 3 clearly shows that TP-GRPO exhibits faster performance improvement under identical conditions.
> >
> > Furthermore, our choice of **rollout size = 8** aligns with common practice in the community. Given these factors, we believe the comparison is fair and adheres to standard experimental protocols.
> >
> >
> >
> > If you have any further questions or would like to continue the discussion, we would be very happy to engage. Thank you again for your thoughtful feedback and for maintaining a positive assessment of our work.

---

### Official Review · Reviewer_ebqn · 2025-10-31

**Soundness:** 3
**Presentation:** 3
**Contribution:** 3
**Rating:** 8
**Confidence:** 4

**Summary:**

This paper begins with an in-depth analysis of the challenges inherent in Generative Process Reward Model (GenPRM) based process evaluation. To mitigate these challenges, the authors propose a novel generative process evaluation mechanism. This mechanism features an intrinsic-signal-driven evaluation (judging reasoning steps based on semantic information) and thought-level, ability-adaptive reward schemes. The mechanism is integrated with the GRPO algorithm to form a new RL algorithm, termed TP-GRPO. Experimental results on mainstream reasoning benchmarks demonstrate that TP-GRPO achieves superior training efficiency and accuracy compared to baselines.

**Strengths:**

1. The paper is well-written and presents the topic clearly.

2. The proposed method is well-motivated and appears technically sound.

3. The empirical evaluation is promising. TP-GRPO is shown to outperform the GRPO baseline and most existing outcome reward-based RLVR methods in terms of both training efficiency and accuracy.

**Weaknesses:**

1. Notable performance gap remains when comparing TP-GRPO to state-of-the-art models with much larger training budgets (e.g., DeepScaler-1.5B-Preview and Skywork-OR1-7B). The paper does present initial scaling trend (i.e., Fig 3), but the experiment appears to be conducted with relatively low training budgets. This limited scope makes it difficult to conclusively determine whether the advantages of TP-GRPO will persist, widen, or saturate as model scale and training budgets increase significantly.

2. The introduction clearly articulates several "design pitfalls" of existing GenPRM-based process evaluation. While the paper does provide some analyses related to these points, the insights are somewhat scattered throughout the experimental section rather than being presented cohesively. The paper would be significantly strengthened if this analysis were consolidated and made more explicit, clearly demonstrating how the proposed method directly mitigates each of the identified pitfalls.

**Questions:**

Overall, this paper is well-written, well-motivated, and the proposed method is technically sound. The empirical results are promising, demonstrating clear improvements over strong baselines, even if they do not surpass all current SOTA methods. In order to maintain my rating, I would like the authors to address the points in the weaknesses above.

---

> ### Author Response · Authors · 2025-11-21
> **Response for Reviewer ebqn (part 1)**
>
> We sincerely appreciate your recognition of the strengths of our work, your understanding of the limitations imposed by our resource constraints, and the time and effort you devoted to reviewing our paper. We are also grateful for your constructive feedback on our writing. In response to your suggestions, we have added additional experiments; however, due to limited time and resources, these experiments are still ongoing. Once the results are available, we will promptly provide feedback. Below, we offer detailed responses to your other questions.
>
>
>
> **Q1: Notable performance gap remains when comparing TP-GRPO to state-of-the-art models with much larger training budgets (e.g., DeepScaler-1.5B-Preview and Skywork-OR1-7B). The paper does present an initial scaling trend (i.e., Fig 3), but the experiment appears to be conducted with relatively low training budgets. This limited scope makes it difficult to conclusively determine whether the advantages of TP-GRPO will persist, widen, or saturate as model scale and training budgets increase significantly.**
>
> **RA1:** We appreciate your understanding of our low-resource setting. Due to hardware limitations, we were unable to train until full convergence. As reported by DeepScaler-1.5B-Preview [1], on-policy GRPO with a batch size of 128 still has not converged after 1000 steps. Under our setting (batch size = 5), this corresponds to roughly 25,600 steps, which is far beyond our computational budget.
>
> We would also like to clarify our primary research objectives: first, to demonstrate that process rewards can enhance training efficiency in the LRM reasoning task; and second, to design a principled generative process reward mechanism that better addresses the limitations of existing GenPRM-based methods. Accordingly, our experimental setup is fully aligned with these research goals.
>
> Importantly, even under limited resources, TP-GRPO demonstrates a clear improvement in training efficiency. For instance, on a 1.5B model, TP-GRPO surpasses the previous state-of-the-art GRPO performance in just 140 training steps. To examine its performance trend (motivated by concerns similar to yours), we continued training TP-GRPO and observed a sustained upward trend (as shown in Figure 3) and stronger performance at 369 steps (avg.@16 of 34.79 in AIME 2024).
> To further investigate the subsequent trend, we have continued training the model. **We sincerely apologize that, due to time and resource constraints, multiple additional experiments are required, and the current experiment are still running. Once it is completed, we will promptly update the results in this rebuttal and look forward to engaging in further in-depth discussion with you.**
>
> Finally, we hope to have the opportunity to discuss with you the future prospects of process rewards for LLM reasoning. In fact, we remain cautious about whether prolonged training with process rewards can ultimately outperform outcome-only rewards. Given the vast state space of complex reasoning and the high cost of process evaluation, allocating substantial resources to "exploitation" risks overfitting to local optima. It may be more beneficial to invest the budget in training on more problems and exposing the model to different solution patterns.
>
> From a research perspective, however, our objective is to investigate the potential value of process rewards in an era primarily shaped by RLVR-style approaches. A possible path may be alternating the use of outcome and process rewards. The model can first be trained with outcome rewards to preserve exploration and reflective behaviors, followed by applying process rewards to high-quality offline trajectories to improve trajectory utilization and strengthen fundamental reasoning. Building on that foundation, continuing to use outcome rewards to drive exploration on new tasks may help combine the strengths of both paradigms. Of course, this remains a preliminary hypothesis requiring further systematic validation. We note that this analysis focuses on correctness-based process rewards; exploration-oriented variants that increase policy entropy or behavioral diversity may yield different conclusions.
>
> We also believe that process rewards may have greater potential value in agent-based environments compared with mathematical reasoning tasks. These settings require continuous interaction with external systems, and rewards are often extremely sparse, particularly when interaction is costly and inefficient, such as in deep research or VLN. Under such circumstances, maximizing the value of each trajectory is crucial, and process rewards may play an essential role in improving sample efficiency and enabling long-horizon learning.
>
> We sincerely welcome any criticism, clarification, or discussion regarding our perspective.

---

> > ### Author Response · Authors · 2025-11-21
> > **Response for Reviewer ebqn (part 2)**
> >
> > **Q2: The introduction clearly articulates several "design pitfalls" of existing GenPRM-based process evaluation. While the paper does provide some analyses related to these points, the insights are somewhat scattered throughout the experimental section rather than being presented cohesively. The paper would be significantly strengthened if this analysis were consolidated and made more explicit, clearly demonstrating how the proposed method directly mitigates each of the identified pitfalls.**
> >
> > **RA2:** Based on your feedback, we first added detailed analyses of the corresponding pitfalls of existing GenPRM-based methods before introducing each method. We then further clarified in the experimental section which empirical results validate each analysis. Below we provide a concise summary:
> >
> > **(1) Evaluation task should be decoupled from reasoning ability**
> >
> > Existing GenPRM-based evaluation methods primarily rely on having the evaluator “think” to identify incorrect steps in the Long CoT. This paradigm implicitly assumes that the GenPRM has sufficient reasoning ability to correctly solve the original problem or its subproblems so that it can detect inconsistencies in the solution. However, this design has three major drawbacks:
> >
> > a. **It substantially raises the capability requirement for the GenPRM**, essentially assuming that the evaluator must be at least as strong as the actor model. As the actor improves, the GenPRM must also scale accordingly.
> >
> > b. **This paradigm heavily depends on the evaluator’s reasoning ability while ignoring internal semantic signals naturally present in the original Long CoT.** Collectively, these issues indicate that existing GenPRM-based evaluation mechanisms are overly reliant on reasoning ability.
> >
> > We therefore argue that evaluation should be decoupled from reasoning and that the missing information from this decoupling should be compensated by leveraging the intrinsic semantic signals already embedded within the Long CoT.
> >
> > To achieve this, Section 3.1.1 introduces our **intrinsic-signal–guided evaluation mechanism**:
> >
> > - For correct solutions, we extract reflection signals to assist evaluation.
> > - For incorrect solutions, we rely mainly on the GenPRM’s semantic matching ability.
> >
> > This mechanism maximizes the use of semantic information inherent in the solution and depends primarily on semantic understanding and matching (foundational capabilities of LLMs) rather than high-level reasoning.
> >
> > **Experimental evidence:**
> >
> > - Table 3 compares TP-GRPO with an LLM-as-a-judge approach that uses the evaluator’s reasoning to assess step correctness. TP-GRPO consistently outperforms these reasoning-based methods.
> > - We further evaluate multiple GenPRMs (Qwen3-32B, Qwen3-4B, and Gemma-3-12B-it) whose reasoning capabilities decrease in order. While TP-GRPO experiences slight degradation, the performance is still better than the LLM-as-a-judge baseline. This demonstrates that TP-GRPO does not strongly depend on evaluator reasoning capability. Additional experiments using LLM-as-a-judge with varying evaluator strengths are ongoing and will be updated as soon as completed.

---

> > > ### Author Response · Authors · 2025-11-21
> > > **Response for Reviewer ebqn (part 3)**
> > >
> > > **(2) Reward density significantly affects policy optimization**
> > >
> > > Existing GenPRM-based methods typically adopt simple step segmentation (e.g., splitting by "\n\n" or letting the evaluator segment directly), resulting in very fine-grained steps. Assigning rewards at such fine granularity can cause optimization misalignment, that is advantage values become inconsistent with the actual correctness of steps.
> > >
> > > We illustrate the potential adverse effects of dense process rewards in Fig 2(b). In the upper bar chart, where only the final outcome reward is provided, all tokens in the solution are uniformly encouraged. In contrast, in the middle bar chart, although the 4th step is correct, the five subsequent incorrect steps yield a negative cumulative return, causing this correct step to be erroneously suppressed during training. To further validate the prevalence and severity of this issue, Fig. 4 and 5 present quantitative analyses from two perspectives: the distribution of token-level advantages and the mutual information between token advantages and correctness labels. The results consistently confirm the existence of this problem.
> > >
> > > To address this, we propose decoupling evaluation granularity from reward granularity. Our step-merging strategy aggregates consecutive steps with consistent correctness labels into “thoughts,” and rewards are assigned at the thought level. Evaluation is still performed at the step level, but rewards are computed over merged thoughts.
> > >
> > > **Experimental evidence:**
> > >
> > > - Figures 4 and 5 show that assigning rewards at the step level leads to (i) large advantage variance across tokens within the same solution—hurting optimization stability—and (ii) low mutual information between token advantages and correctness—indicating mismatched optimization direction.
> > > - In Section 4.3.1, removing step-merging causes significant degradation (AIME24: 33.12 → 31.66; AIME25: 25.63 → 22.29; AMC23: 64.01 → 62.19), demonstrating the necessity of step-merging.
> > >
> > >
> > >
> > > **(3) Process rewards need to balance exploration and exploitation**
> > >
> > > Correctness-based process rewards assign negative signals to erroneous steps. While this can reduce harmful behaviors and accelerate convergence to correct solutions, it may also suppress the initiative of trial-and-error exploration. Therefore, the use of process rewards should take into account reducing the inhibition on the exploration ability. To this end, we propose an **ability-adaptive reward mechanism**:
> > >
> > > - For **harder problems**, exploration should be prioritized, so the process reward is down-weighted.
> > > - For **easier problems**, exploitation should dominate, so the process reward is up-weighted to better leverage useful parts of the solution.
> > >
> > > **Experimental evidence:**
> > >
> > > Section 4.3.1 shows that replacing adaptive rewards with fixed ±1 rewards (Table 3, w/o Stage II/S2) leads to clear performance drops, confirming the necessity of ability-adaptive reward computing.
> > >
> > >
> > > Finally, we sincerely thank you again for your recognition and suggestions. We will update the experimental results here once the experiments are complete and look forward to engaging in further in-depth discussion with you.

---

> > > > ### Author Response · Authors · 2025-11-24
> > > > **Response for Reviewer ebqn (part 4)**
> > > >
> > > > **Supplementary experiments for Q1**
> > > >
> > > > Due to limitations in TRL’s checkpoint-saving mechanism, we were unable to continue training from the old experiment in Figure 3 and could only evaluate using the existing checkpoints. To address Reviewer N8ji’s request regarding randomness, we conducted additional training and analysis following robustness experiments with different random seeds.
> > > >
> > > > In this experiment, the model was first trained to step 152, achieving a Pass@16 score of 33.10 on AIME24. We then trained for 15 additional checkpoints, reaching 33.33 at step 359, indicating that training with process rewards had essentially converged. We attribute this to rapid convergence potentially causing the model to fall into local optima: while process rewards improve training efficiency, they may also suppress broader exploratory behavior.
> > > >
> > > > To validate this hypothesis, we continue training from the 152-step checkpoint using only outcome rewards for an additional 450 steps, reaching a higher 34.79 Pass@16. This demonstrates that even after convergence under process rewards, switching to outcome rewards can further improve performance, albeit requiring more training steps due to lower trajectory utilization efficiency of outcome rewards.
> > > >
> > > > Based on these observations, we plan to explore a hybrid alternating training paradigm in the future: initially using outcome rewards to promote broad exploration and diversify experience; then applying process rewards for intensive exploitation to accelerate convergence; and finally returning to outcome rewards to re-expand the exploration space after stabilization. This strategy aligns with human learning: completing tasks first, then systematically reflecting to distill valuable experience, and subsequently applying accumulated knowledge to the next task.
> > > >
> > > > We sincerely welcome any criticism, clarification, or discussion regarding our viewpoint.

---

> > > > > ### Comment · Reviewer_ebqn · 2025-11-27
> > > > >
> > > > > Thank you for the clarifications. My concerns have been addressed by the point-by-point rebuttals and I will maintain my score to reflect the hard-works being done.

---

> > > > > > ### Author Response · Authors · 2025-11-27
> > > > > >
> > > > > > We would like to once again express our sincere appreciation for your thoughtful review comments. Thank you for taking the time to review our clarifications and for maintaining your positive assessment. We are truly grateful for your recognition of our efforts.

---

### Comment · Area_Chair_824s · 2025-11-27
**Please respond to authors**

Dear Reviewer iXx8, Reviewer LpBz, and Reviewer N8ji ,

Could you reply to authors' initial responses as soon as possible? If you have any further questions or concerns, please raise them.

Thanks

AC

---

> ### Author Response · Authors · 2025-11-28
>
> We sincerely appreciate your help in contacting reviewers, and thank you for your time and coordination!

---

### Author Response · Authors · 2025-11-29
**Rebuttal Summary**

We sincerely thank all reviewers for their thoughtful feedback and constructive suggestions. In our detailed responses, we have endeavored to address all major concerns through additional clarifications, theoretical discussions, and supplementary experiments. We also made targeted revisions to the manuscript.

After discussion  and AC’s kind reminder, all reviewers indicated that our responses largely resolved their concerns. **Notably, two reviewers  who initially gave negative scores, N8ji and LpBz, expressed their decision to raise scores. And the two reviewers give initial positive scores, ebqn and iXx8, confirmed their scores.** We greatly appreciate all reviewers’ recognition of our work.

Below, we summarize the key strengths highlighted by the reviewers and our responses to the main concerns.


> **STRENGTHS**

We thank the reviewers for highlighting the key strengths of our work:

- **Well-motivated and clearly defined problem formulation (Reviewer ebqn, iXx8, N8ji, LpBz):** All reviewers agree that this paper clearly articulates the motivation and pinpoints the fundamental weaknesses of current GenPRM designs, offering insights that help clarify promising directions for future research.
- **Substantive methodological innovation (Reviewer iXx8, N8ji, ebqn, LpBz):** All reviewers highlight the paper’s technical innovations and acknowledge the methodology technically sound and conceptually insightful.
- **Clear writing and well-structured presentation (Reviewer ebqn, LpBz):** Reviewers note that the paper is clearly written, well organized, and easy to follow.
- **Strong empirical performance (Reviewer ebqn):** Experiments show that TP-GRPO consistently outperforms GRPO and most outcome-reward RLVR baselines in both training efficiency and accuracy.



> **Response for MAIN CONCERNS**

We have carefully addressed the main concerns raised by the reviewers:

**(1) Additional baselines (Reviewer N8ji, LpBz)**

We added two baselines: LLM-as-a-Judge and a specially pretrained GenPRM from related work. and systematically compared our method with LLM-as-a-Judge across different base models. These results are included in Sections 4.2 and 4.3.3.

**(2) Fairness of comparisons (Reviewer iXx8, N8ji)**

We clarified that all comparisons with vanilla GRPO were conducted under identical computational settings. In addition to $Effic.$ metric, training curves prove the efficiency of our method under the same budget.

**(3) Extended training (Reviewer ebqn)**

We increased training steps and summarize the resulting trends. We also explored alternating process and outcome reward training, yielding additional insights shared in our response.

**(4) Method details and theoretical foundation (Reviewer iXx8, LpBz)**

We provided a very thorough and detailed explanation to clarify the theoretical foundations behind our "Intrinsic-signal-driven process evaluation", summarized as following:

- Using reflection signals to locate incorrect steps in the reasoning chain is feasible because the reasoning chain is **self-contained** if the final answer is correct, ensuring consistency between trial-and-error and reflection;

- Semantic-based evaluation of erroneous reasoning steps aims to balance reducing dependence on GenPRM capabilities and avoiding over-penalization from outcome-only rewards.

Additionally, we systematically clarified RL-related concepts and our advantage function design, supported by specific references.

**(5) Clearer presentation of contributions and evidence (Reviewer ebqn)**

We added descriptions of corresponding pitfalls before each method component and guiding statements at the start of experiments to clearly link “problem — method — experiment.”

**(6) Differences from related work (Reviewer N8ji)**

Unlike StepWiser, which follows the traditional paradigm and enhances GenPRM capabilities, our method leverages semantic signals from the reasoning process itself, reducing reliance on GenPRM. Therefore, we explore a new methodology rather than extending existing paradigms.

---

> ### Author Response · Authors · 2025-11-30
> **Reviewer Final Feedback Summary**
>
> For clarity, we summarize all reviewers’ final feedback and score updates. All reviewers confirmed that our rebuttal successfully resolved their major concerns, and two reviewers decided to raise their scores accordingly.
>
> - **Reviewer ebqn — Rating: 8, Confidence: 4**
>    *“Thank you for the clarifications. My concerns have been addressed by the point-by-point rebuttals, and I will maintain my score to reflect the hard work being done.”*
> - **Reviewer iXx8 — Rating: 6, Confidence: 4**
>    *“Taking all these points into consideration, I am inclined to maintain my score of 6, which I consider to be slightly positive.”*
> - **Reviewer N8ji — Rating: 4 → 6, Confidence: 4**
>    *“Thanks for the detailed explanation. This addresses my concerns, and I would like to raise my score to 6.”*
> - **Reviewer LpBz — Rating: 2 → ≥4, Confidence: 3**
>    *“I appreciate the authors' detailed responses, which have addressed most of my concerns. I plan to update my score as soon as the system’s edit button is available.”*

---

> > ### Author Response · Authors · 2025-12-02
> > **Clarifications on Initially Negative Scores**
> >
> > To clarify the two initially negative scores, which stem largely from misunderstandings rather than substantive flaws, we provide the following focused explanations:
> >
> > - **Reviewer N8ji:** Although the initial rating was 4, **the reviewer gave good scores for both Soundness and Contribution**. Their concerns focused on the terms in the title, the distinction from related work, and the fairness of comparisons, with the latter two stemming from misunderstandings. **The reviewer also noted that he/she would be happy to raise the score if these concerns were resolved**. We addressed each point in detail to clarify the misunderstandings, after which the reviewer **agreed to raise the rating to 6**.
> >
> > - **Reviewer LpBz:** Although the initial Rating was 2, **the reviewer’s concerns stemmed primarily from confusion and misunderstandings due to our insufficient clarity in theoretical rationale for assumptions and viewpoints; in fact, the reviewer also noted the soundness of our approach in the Strengths section.** These uncertainties led the reviewer to question the foundations of our assumptions, resulting in the somewhat harsh initial score. In our rebuttal, we provided a thorough clarification of the underlying principles and proved the method’s theoretical soundness. This resolved the reviewer’s concerns, and the reviewer **agreed to raise the score**.

---

> > > ### Author Response · Authors · 2025-12-03
> > > **Key Technical Contributions of our Work**
> > >
> > > - **Formalizing the core limitations of GenPRM methods.** This work is the first to discuss the inherent limitations in existing GenPRM methods and formalize three core issues: (1) excessive reliance on GenPRM’s own reasoning capabilities, which raises the training threshold and requires continual GenPRM updates; (2) optimization bias induced by dense reward signals; (3) suppressed exploration caused by inappropriate process rewards. We intend to facilitate broader exploration of GenPRM in future work.
> > > - **Methodological innovations.**
> > >   - Unlike existing methods that rely solely on GenPRM, we first introduce a new generative process evaluation paradigm that leverages **intrinsic signals** emerging in reasoning traces. This design enables effective process evaluation with only basic semantic understanding, substantially reducing dependence on high-level reasoning capabilities.
> > >   - We propose the concept of **thought** rewarding units, created by merging consecutive steps with identical correctness. We are also the first to decouple evaluation granularity from reward granularity, effectively mitigating optimization drift caused by dense rewards.
> > >   - We develop an **ability-adaptive process reward mechanism** that adjusts reward strength based on the model’s current capacity. This reward design also mitigates reward hacking via keeping key-step optimization on the correct direction.
> > > - **Consistent performance and efficiency gains.** Our approach yields stable performance improvements on both 1.5B and 7B models. Moreover, compared with leading baselines, our method delivers **substantial training efficiency gains**, achieving up to **7.83×** and **4.17×** acceleration, respectively.

---

### Meta-Review · Area_Chair_fkoC · 2026-01-06

**Summary:**

The paper studies if reward models (GenPRMs) can accelerate RL training of LRMs by improving the utilization of reasoning trajectories. The method aims to enhance the evaluation of generative models by dynamically adjusting rewards according to the model's performance capabilities. Initial reviews were mixed: 2,4,6,8. As a common concern, reviewers mentioned the need for detailed clarifications regarding the implementation of the intrinsic signals and the adaptive nature of the reward mechanisms. The rebuttal process when in an engaged form and resulted in multiple score increases. As a result, the scores would likely be 8, 6, 6, 2-4 pushing the paper above the acceptance threshold of the conference.

**Reviewer Concerns:**

Reviewers and authors engaged into the discussion during the rebuttal. Specifically, the authors' responses resolved questions about the experimental setup and the differentiation of their reward shaping from standard approaches. There are some concerns left on the scalability to larger models. According to the discussion and comments, multiple reviews considered increasing the score.

**Reviewer Scores:**

ebqn: 8 → 8
iXx8: 6 → 6
N8ji: 4 → 6
LpBz: 2 → 2-4

---

### Decision · Program_Chairs · 2026-01-26

Accept (Poster)